# An explainable model of host genetic interactions linked to COVID-19 severity

Anthony Onoja[1], Nicola Picchiotti[2,3], Chiara Fallerini[4,5], Margherita Baldassarri[4,5], Francesca Fava[4,5,6], GEN-COVID Multicenter Study*, Francesca Colombo [7], Francesca Chiaromonte[8,9], Alessandra Renieri [4,5,6✉], Simone Furini[4] & Francesco Raimondi [1✉]

We employed a multifaceted computational strategy to identify the genetic factors contributing to increased risk of severe COVID-19 infection from a Whole Exome Sequencing (WES) dataset of a cohort of 2000 Italian patients. We coupled a stratified $k$-fold screening, to rank variants more associated with severity, with the training of multiple supervised classifiers, to predict severity based on screened features. Feature importance analysis from tree-based models allowed us to identify 16 variants with the highest support which, together with age and gender covariates, were found to be most predictive of COVID-19 severity. When tested on a follow-up cohort, our ensemble of models predicted severity with high accuracy (ACC = 81.88%; AUCROC = 96%; MCC = 61.55%). Our model recapitulated a vast literature of emerging molecular mechanisms and genetic factors linked to COVID-19 response and extends previous landmark Genome-Wide Association Studies (GWAS). It revealed a network of interplaying genetic signatures converging on established immune system and inflammatory processes linked to viral infection response. It also identified additional processes cross-talking with immune pathways, such as GPCR signaling, which might offer additional opportunities for therapeutic intervention and patient stratification. Publicly available PheWAS datasets revealed that several variants were significantly associated with phenotypic traits such as "Respiratory or thoracic disease", supporting their link with COVID-19 severity outcome.

[1] Laboratorio di Biologia Bio@SNS, Scuola Normale Superiore, Pisa, Italy. [2] University of Siena, DIISM-SAILAB, Siena, Italy. [3] Department of Mathematics, University of Pavia, Pavia, Italy. [4] Med Biotech Hub and Competence Center, Department of Medical Biotechnologies, University of Siena, Siena, Italy. [5] Medical Genetics, University of Siena, Siena, Italy. [6] Genetica Medica, Azienda Ospedaliero-Universitaria Senese, Siena, Italy. [7] Istituto di Tecnologie Biomediche—Consiglio Nazionale delle Ricerche, Segrate, MI, Italy. [8] Dept. of Statistics and Huck Institutes of the Life Sciences, Penn State University, University Park, PA 16802, USA. [9] Institute of Economics and EMbeDS, Sant'Anna School of Advanced Studies, 56127 Pisa, Italy. *A list of authors and their affiliations appears at the end of the paper. ✉email: alessandra.renieri@unisi.it; francesco.raimondi@sns.it

The coronavirus disease 2019 (COVID-19) pandemic, caused by the infection with severe acute respiratory syndrome coronavirus 2 (SARS-CoV-2), is challenging health, economical and societal systems worldwide at an unprecedented level. The SARS-CoV-2 infection is characterized by a large variation in consequence ranging from asymptomatic to life-threatening conditions such as viral pneumonia and acute respiratory distress syndrome (ARDS). ARDS is caused by an exaggerated host immune response leading to lung injury, which starts at the epithelial–interstitium–endothelial interface with increased vascular permeability and extravasation of immune cells, mostly macrophages, and granulocytes. Infected epithelial cells and debris bind immune cell receptors, triggering the release of inflammatory cytokines (predominantly IL-6, IL-1, and TNF-α) and activating fibroblasts, resulting in a cytokine release syndrome[1].

Established host risk factors for disease severity, such as increasing age, male gender, and higher body mass index[2], do not explain all the variability in disease severity observed across individuals. Genetic factors contributing to COVID-19 susceptibility and severity may provide novel biological insights into disease pathogenesis mechanisms, new drug targets as well as new means for patient stratification. It is important to consider that, despite the recent development of vaccines, treating the disease remains an important goal in the clinics. The first genetic factors described to contribute to COVID-19 severity were rare loss-of-function variants in genes involved in type I interferon (IFN) responses[3–7]. At the same time, several GWAS projects investigating the contribution of common genetic variation[8,9] to COVID-19 have provided robust support for the involvement of various genomic loci associated with COVID-19 severity and susceptibility, with the strongest finding for severity being located on chromosome 3. Until now, the Italian GEN-COVID Multicenter Study contributed to the identification of rare variants[6,10] and common polymorphisms[11–13] associated with COVID-19 severity through the collection of more than two thousand biospecimens and clinical data from SARS-CoV-2-positive individuals[14] and whole exome sequencing (WES) analysis. The COVID-19 Host Genetics Initiative (COVID-19 HGI) (https://www.COVID-19hg.org) has recently provided the most comprehensive picture of host genetic factors linked to COVID-19 severity through meta-analyses of tens of studies from 19 countries[15].

While GWAS studies provide solid evidence of the host genetic factors individually associated with COVID-19 severity, they most often fail to provide an organic picture about their interplay. By learning (non-)linear patterns from data in a human interpretable fashion, explainable machine learning algorithms might help in understanding the multifactorial nature of the interactions between host genetics and COVID-19, at the same time providing effective tools for risk and severity forecasting.

In 2020, the Italian GEN-COVID Multicenter Study started to investigate how the combination of common and rare variants could determine COVID-19 severity in a pilot study including WES data of a first small cohort of hospitalized patients[16]. Previous and ongoing efforts entailed machine learning techniques (i.e. LASSO logistic regression models) in combination with a boolean representation of genetic variants to identify the most informative features associated with the severity which were used to compile an Integrated PolyGenic Score for COVID-19 severity predictions[17,18]. In this study, we combined variant case-control screening, supervised binary classifiers training, feature importance analysis, and dimensionality reduction techniques with pathway enrichment and phenotype association studies to identify a few dozens genetic variants contributing to increased risk of severe COVID-19 infection from a Whole Exome Sequencing (WES) dataset of a cohort of Italian patients.

## Results

**Comparing genetic variation in severe and asymptomatic individuals**. We considered the Whole Exome Sequencing (WES) dataset of germline variants from 1982 European descent patients provided by the GEN-COVID Multicenter Study group[14]. All subjects were classified according to the grading scheme by the World Health Organization (WHO), refined based on an ordinal logistic model using age as input feature for sex-stratified patients[17]. Demographic (sex, age, and ethnicity) and clinical data (family history, pre-existing chronic conditions, and SARS-CoV-2 related symptoms) were also collected (Fig. 1a; see "Methods").

We started our analysis from a total of 1.057 M simple variants which were screened to identify mutations associated with severe patients, likely representing risk factors, from those associated with asymptomatic patients, more likely contributing to protection. We employed log odds ratio statistics, using an additive model, to screen variants significantly associated with either severe or asymptomatic groups (Fig. 1a, b; see "Methods"). We performed the screening on the majority portion (training set) of a randomly split dataset (keeping 80% of the samples for training and 20% for testing), to find a set of variants to be used as features set for downstream ML and pathways analysis. To ensure robustness, we repeated the splitting procedure five times, employing a stratified five fold cross-validation scheme, by performing the screening on the training set and finally retaining those variants found to be significantly enriched in each of the five splits (Fig. 1d; see "Methods"). We found on average 1130 variants significantly enriched across the five folds (Data S1).

**Genetic variants predict severity through supervised ML classifiers**. We embedded the stratified five fold screening within a supervised classifier training procedure (Fig. 1d; see "Methods"). For each random split of the dataset, we trained the model by considering the variants screened in the training set (80% of the original dataset), and tested it on the corresponding held-out portion (20% of the dataset) of the same split. For each screened random split, we trained multiple models using a stratified five fold Cross-validation (five fold CV) grid search to estimate optimal hyperparameters for supervised classifiers training (Fig. 1d; see "Methods"). XGBoost was the algorithm that displayed the smallest drops between training and testing accuracies, achieving the best average performance during testing across the five folds (Fig. 2a; Data S2). In more details, the best XGBoost model had the following performances: Precision=77.27%, Recall=83.33%, MCC = 46.69%, AUCROC = 80, Accuracy=75%, F1 = 80.2% (Data S2).

Overall, we found that 3217 unique variants (out of a total of 3258 unique, screened variants), corresponding to 2546 unique genes, had non-zero coefficients in at least one of the five, tree-based models (i.e. RF or XGBoost). However, the XGBoost classifier led to a sharper reduction of relevant variants (1086, corresponding to 1049 genes, with non-zero feature importance in at least one model), consisting of a subset of those identified with the RF models. As expected, clinical covariates such as age and gender were found among the features with the highest median of the distribution of importance coefficients collected from XGBoost models (Fig. 2b). Among this shortlist, only 16 variants (and corresponding genes; Data S1) consistently received non-zero coefficients in all tree-based models, out of which 9 variants were found to be enriched among severe patients (Fig. 2b, c). To confirm the predictive performance of these variants, we re-trained the models by considering only this subset of variants, plus age and gender covariates, and we calculated aggregated performances by considering the median of the probabilities outputted by each model for each sample in the testing set (see "Methods"). While age and gender covariates

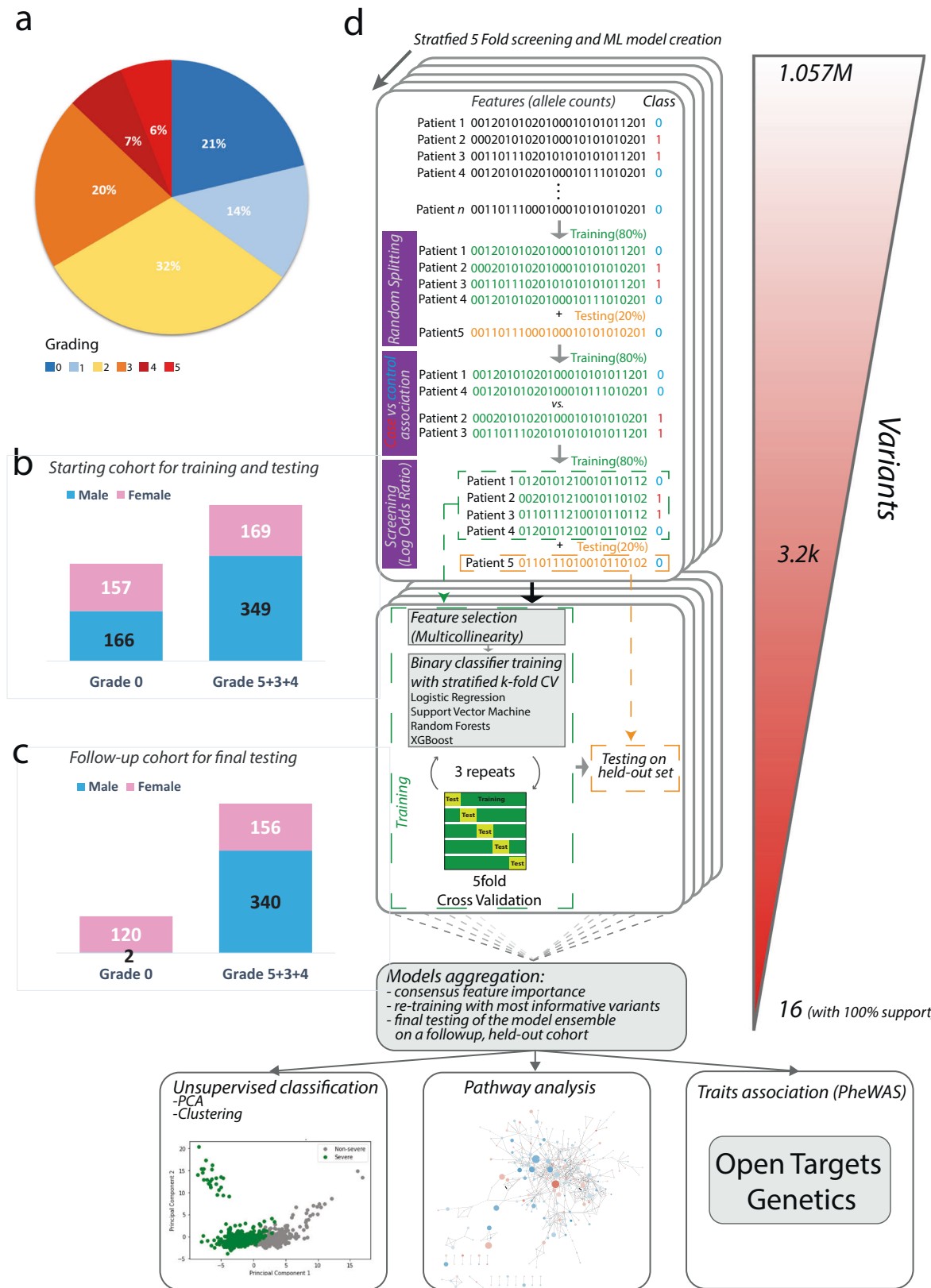

**Fig. 1 Patient cohort and workflow of the computational pipeline. a** piechart with the fraction of sequenced patients for each grading group; **b** stacked bar-charts with distribution of patients in the two groups (severe=5 + 4 + 3; asymptomatic=0), and their gender composition, whose variants were used for screening, training and initial testing; **c** stacked bar-charts with distribution of patients in the two groups (severe=5 + 4 + 3; asymptomatic=0), and their gender composition, from a follow-up cohort used for final testing of the model; **d** workflow of the bottom-up computational strategy to identify and interpret variants linked to COVID-19 severity.

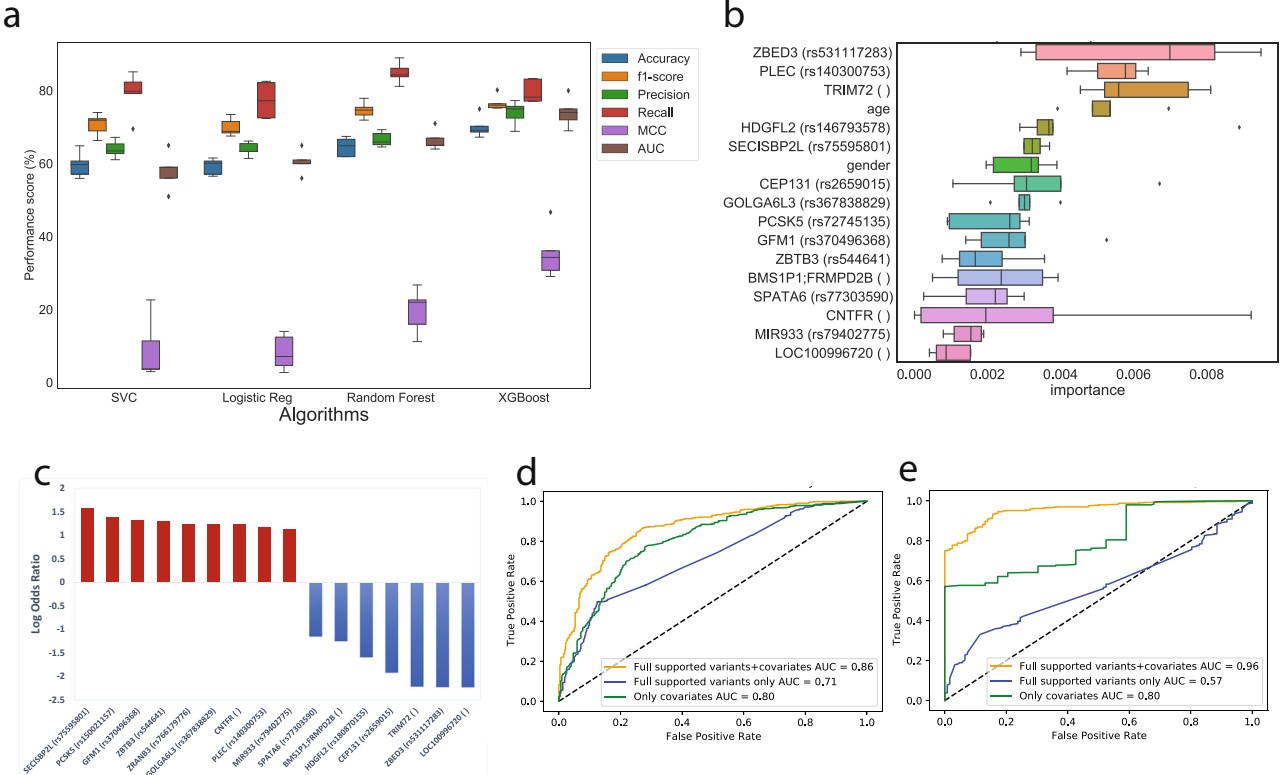

**Fig. 2 Performances of the supervised classifier for prediction of COVID-19 severity. a** Distribution of performance metrics of different algorithms during testing on the five folds. The horizontal line inside each box represents the median value, and the height (whiskers) of each of the boxes depict the standard error (variability) of a particular performance metrics under consideration as scored across the five fold CVs by the employed supervised ML algorithms. The dotted points above and below the individual box-and-whisker lines are potential outliers that are above or below the 25th percentile, and the 75th percentile; **b** feature importance distribution for features with non-zero importance across the five folds. The characteristics of each box-plot are as in Fig. 2a; **c** log-odds ratio of the 16 variants with full support in XGBoost trained models; **d** performances of the predictors with 16 variants plus covariates (age and gender; orange), only co-variates (green), all screened variants plus covariates (blue) in the held-out test set (samples $n = 168$); **e** performances of the predictors with 16 variants plus covariates (age and gender; orange), only co-variates (green), all screened variants plus covariates (blue) in a follow-up testing set cohort (new samples $n = 618$).

alone retained high predictive power (AUCROC = 80%), the addition of these most informative genetic features led to an increase of performances (AUCROC = 86%, best model AUCROC = 91%; Fig. 2d; Data S3).

We observed a high level of performance when we tested the ensemble of models trained with only informative variants on a follow-up cohort of 618 individuals (122 asymptomatic, 496 severe; Fig. 1c), either at the individual model level or at the ensemble one (Data S4). In fact, when computing aggregated metrics by considering the median of the probability distribution collected from the ensemble of models (Data S4, 5; see "Methods"), we identified severe patients with good accuracy (ACC = 81.88%; AUCROC = 96%), performing considerably better than the ones obtained by training with only covariates or variants (Fig. 2e; Data S4). The model also showed good performances on an additional validation set comprising a total of 375 samples excluded from both training and testing due to inconsistent classification from the WHO grading and the ordinal logistic model adjusted by age (ACC = 85.34%, MCC = 67.8%, AUCROC = 91.4%; Fig. S1; Data S6).

**Risk and protective genetic factors impinge on modular, interconnected networks underlying distinct biological processes.** We analyzed the subset of variants receiving non-zero feature importance in at least one XGBoost model to provide a

mechanistic explanation for their potential interaction with COVID-19 infection. We performed pathway analysis by mapping mutated genes in a functional interaction (FI) network (i.e., Reactome FI network; see Methods). We built a general FI network (Fig. 3b), as well as networks specific for clinical groups, by grouping variants and genes enriched in severe and asymptomatic patients (Fig. 3a). Pathway analysis on group-specific networks revealed patterns of significantly enriched processes connected to either risk or protection (Fig. 3a).

In severe patients we found significant processes associated with cardiomyopathies, e.g. *Arrhythmogenic right ventricular cardiomyopathy* (FDR = 4.03E–05), *Calcium signaling pathways* (FDR = 4.22E–02), and immune response such as C-type leptin receptors (CLRs) (FDR = 5.67E–02) (Fig. 3a; Data S7). Asymptomatic patients were instead characterized by distinct processes, including *Fanconi anemia pathway* (FDR = 7.89E–04) and DNA repair processes such as *HDR through HRR or SSA* (FDR = 4.84E–03), (Fig. 3a; Data S8).

The general FI network comprised a total of 344 mutated genes and 630 functional interactions, marking a high degree of interconnection between affected genes, which participate in different, cross-talking biological processes. Cluster analysis on the general FI network revealed modules characterized by specific pathways. Intriguingly, we found out that no cluster exclusively contained variants enriched in severe or asymptomatic patients. In detail, the largest cluster (i.e. Module 1; 43 nodes) encompassed

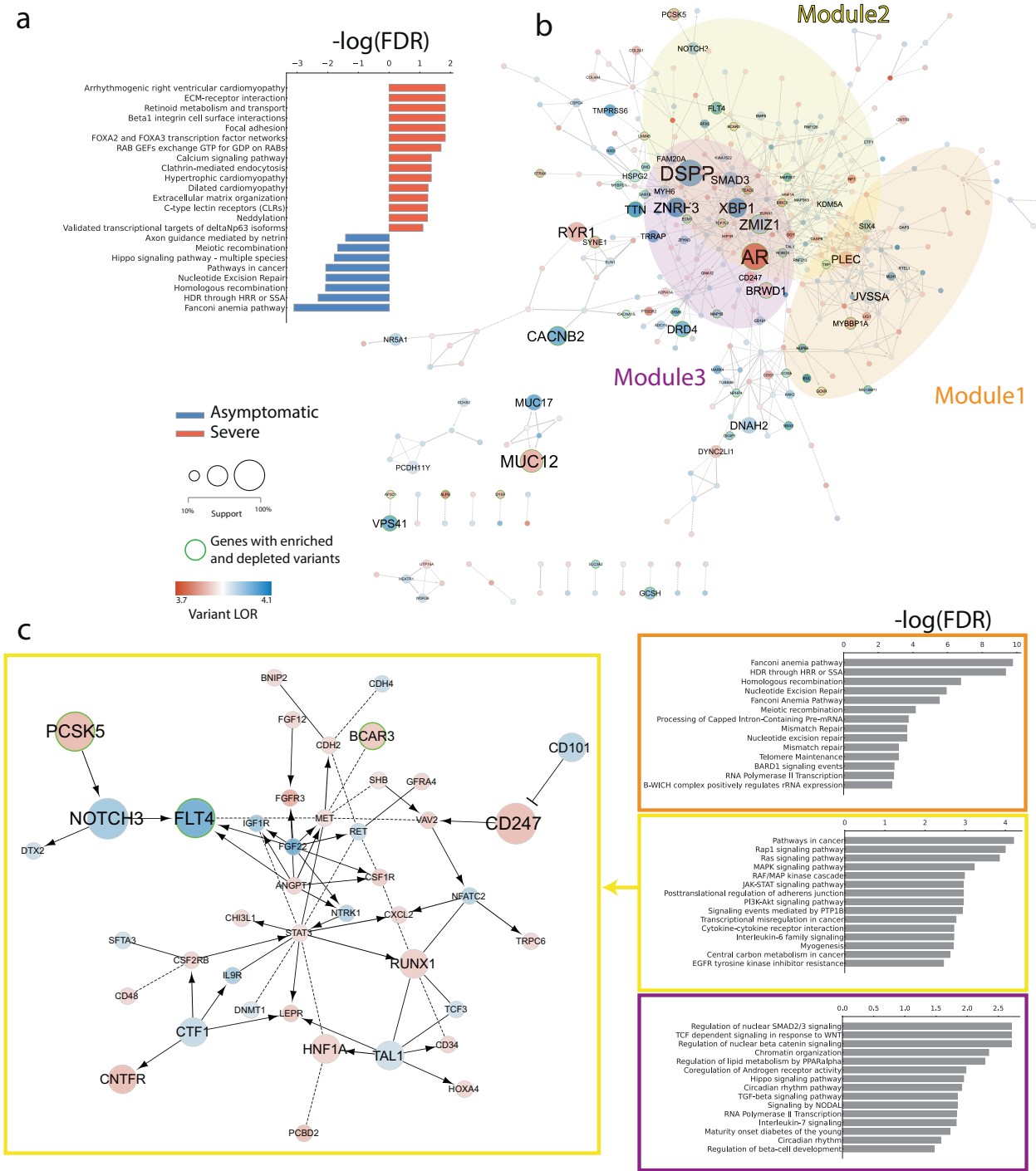

**Fig. 3 network analysis and pathway enrichment. a** Pathways overrepresented among variants with non-zero feature in at least one XGB model and enriched in either severe(red) or asymptomatic (blue); **b** reactome FI network of genes affected by variants with non-zero feature importance from XGBoost. Node diameter is proportional to the number of variants with non-zero coefficients in any tree based models. Node color is instead proportional to the LOR with the highest absolute value among the variants associated to a given gene. The top 3 modules identified within the network are highlighted and corresponding enriched processes displayed as barcharts colored with cluster specific corresponding colors; **c** FI network zoomed representation of the 2nd largest cluster.

*Fanconi anemia pathway* (FDR = 2.46E–07) and DNA repair processes such as *HDR through HRR or SSA* (FDR = 4.51E–06) or *Homologous recombination* (FDR = 1.76E–03) (Fig. 3b). In this cluster, we found that the gene characterized by the variant with the strongest model support (ms) (i.e. fraction of tree-based models assigning non-zero feature importance; see Methods) is *MYBBP1A*

rs117615621, which is enriched in asymptomatic patients (Odds Ratio OR) = 0.26; *p* value = 6.5E–03; ms = 90%; (Data S1, S8).

The second-largest module (Module 2; 42 nodes) involves genes mediating signal transduction cascades such as Ras GTPases, e.g. Rap1 signaling pathway (FDR = 1.01E–04) or MAP kinases, e.g. MAPK signaling pathway (FDR = 5.95E–04)

(Fig. 3b, c). We also found processes more directly linked to the immune and inflammatory response to the virus, such as the JAK-STAT signaling pathway (FDR = 1.11E-03), Cytokine-cytokine receptor interaction (FDR = 1.92E–03), and Interleukin-6 family signaling (FDR = 1.92E–03) (Fig. 3b, c). All three pathways include the *CNTFR* gene, which codes for the alpha subunit of the receptor for the ciliary neurotrophic factor, and is affected by a novel variant (chr9:34557898:A: T) enriched in severe patients (lor=1.230663067; *p* val=2.2E–04; Data S1). Intriguingly this variant was ranked in the top 20 genes with the highest median importance (Fig. 2b) and received 100% model support (Fig. 2c). Another variant with 100% support within the same cluster is rs150021157, which is significantly enriched among severe patients (OR = 3.95; *p* val=1.9E–03; Data S1,S8), and it affects the *PCSK5* gene, a serine endoprotease which processes various proteins including cytokines, *NGF*, renin and which has been reported to regulate the viral life cycle[19].

The third-largest module (Module 3; 38 nodes) is characterized by the *Regulation of nuclear SMAD2/3 signaling* pathway (FDR = 1.95E–03) as the most enriched pathway, therefore being tightly interconnected with cluster 2. The variant *SMAD3* rs897912452 (OR = 0.31; *p* val=5.1E–4) and the novel *ZMIZ1* 10:79307376:-:GGGGGGGGGG (OR = 0.27; *p* val=6.18E–05) have the highest support (ms=90%) and are found enriched in asymptomatic patients. Additionally, the latter gene *ZMIZ1* participates in another significant pathway, *Coregulation of Androgen receptor activity* (FDR = 0.01), which also entails *AR*, which carries several mutations which, depending on the specific genic locus, can be found enriched either in severe or asymptomatic patients with variable support (Fig. 3, S2; Data S1, S8).

We found additional potentially relevant pathways in the remaining modules. Module 4 (33 nodes) contains genes involved in Deubiquitination (FDR = 1.15E-05), a process frequently modified by viral infection[20], as well as several other pathways mediating innate immune response such as the TNF receptor signaling pathway (FDR = 1.15E-05) (Fig. S3; Data S9). Within this module we found the *PLEC* gene, affected by the variant rs140300753 (OR = 3.2, *p* val=2.8E-03, ms=100%), which is enriched in severity and received 100% support from tree-based models (Data S1).

In the remaining clusters we found additional processes with high translational and therapeutic potential. For instance, we found several GPCR-signaling instances significantly enriched in Modules 6 (e.g. *G alpha (i) signaling events*, FDR = 3.69E–04) and 8, which exclusively entails GPCR-downstream signaling pathways and where again the *G alpha (i) signaling events* (FDR = 2.56E–09) and *G alpha (q) signaling events* (FDR = 4.83E–08) are the two downstream pathways most significantly over-represented (Fig. S4; Data S9).

We also found that a few genes whose variants have been identified through our pipeline are among the ones carrying top associations to severity as assessed from studies of the COVID-19 HGI (https://app.COVID-19hg.org/variants)[15]. In detail, variants of 9 out of the 43 genes identified from GWAS studies are also present in our list, including: *ABO, ARL17A, ARL17B, DPP9, LRRC37A, LRRC37A2, RAVER1, TMEM65, ZBTB11* (Data S1).

**Severe patients tend to cluster together using only more informative variants**. We applied unsupervised clustering and dimensionality reduction techniques (i.e. Principal Component Analysis (PCA)) to group patients based on the genetic distance calculated by considering the most informative variants selected after screening and supervised machine learning procedure. By projecting the patients on the first two PCs followed by *k-means* clustering (see "Methods"), we detected three groups of patients

in the original cohort (Fig. 4a–c). The two largest clusters were separated by PC1. The one, 515 patients, was characterized by a majority of severe cases (78% of the total). The second cluster was instead characterized by a prevalence of asymptomatic patients (70% of the total). Finally, a third small cluster was identified through the combined usage of PC1 and PC2 and it was characterized almost exclusively by severe patients (95% of 24 patients in total). Notably, the severity of this cluster is only partially explained on the basis of either gender (59% males and 37% females; Fig. 4c) or age (Fig. S5a). This cluster was characterized by peculiar genetic features, with a smaller number of variants and a neat prevalence of risk over protection factors (Fig. S5b). Remarkably, a total of 7 (out of 9 overall enriched in severe patients) variants with 100% support from XGB models were also found in this cluster (Data S10). Network analysis of the mutated genes in this predominantly severe cluster highlighted several common processes as well as candidates for drug targeting. In particular, several GPCRs (*ADRB2, ADRA1, GRM6*), ion channels (*GRIN1, CACNA1G*), (receptor tyrosine) kinases (*NTRK1, CSF1R, GAK*) and nuclear hormone receptors (*AR, THRB*) participate to this network and can be readily targeted by approved drugs (Fig. 4c; Data S11).

**Important variants are associated with disease traits linked to COVID-19 severe phenotypes**. To provide further evidence of a functional relationships between our variants and COVID-19 severe phenotypes, we checked available open-access integrative resources (i.e. Open Target Genetics initiative[21]) which aggregate human GWAS and functional genomics data to link between GWAS-associated loci, variants, and likely causal genes. In particular, we considered Phenome Wide Association Study (PheWAS) analysis considering a wide range of diseases and traits to identify the phenotypes associated with our variants (see Methods). Intriguingly, we found that many identified variants are associated with traits or phenotypes which might be linked with either risk or protection from severe consequences to the viral infection.

For example, by considering variants with non-zero importance in at least one XGB model, we found that those enriched in severe patients were 70% of the total associated with the category "respiratory or thoracic diseases" (see Fig. 5A). Among the specific traits with strong associations to more supported variants, we found instances such as "Doctor diagnosed emphysema" (*ITPKA*, rs41277684; *LTK*, rs35932273), the latter variant associated also to "Other alveolar and parietoalveolar pneumopathy", "Respiratory disorders in diseases classified elsewhere" (*KCNB1*, rs34467662), "Chronic bronchitis/emphysema" (*C12orf43;HNF1A*, rs11065390; *SLC47A2*, rs34399035), "Acute sinusitis" (*SHANK2*, rs146204677), "Pleural plaque" (*CFAP74*, rs141833643), "Allergic asthma" (*SYTL2*, rs61740616 and rs35751209), "Symptoms and signs involving the circulatory and respiratory systems" (*PCSK5*, rs150021157) (Fig. 5b). Although more weakly associated and supported by our models, we also found several associations with chronic obstructive pulmonary disease (COPD) both in "respiratory or thoracic diseases" and in "infectious disease" categories (Data S12). Other disease categories displaying a net prevalence of phenotypic associations for variants enriched among severe were "immune system disease", with multiple variants associated with specific traits such as "Autoimmune diseases" "Immunodeficiency with predominantly antibody defects" or "Noninfectious disorders of lymphatic channels", and "pancreatic disease" (Fig. 5a; Data S12).

Two of the variants enriched among severe patients which had highest importance in all our models (i.e. *PCSK5* rs150021157 and PLEC rs140300753) were significantly associated with the

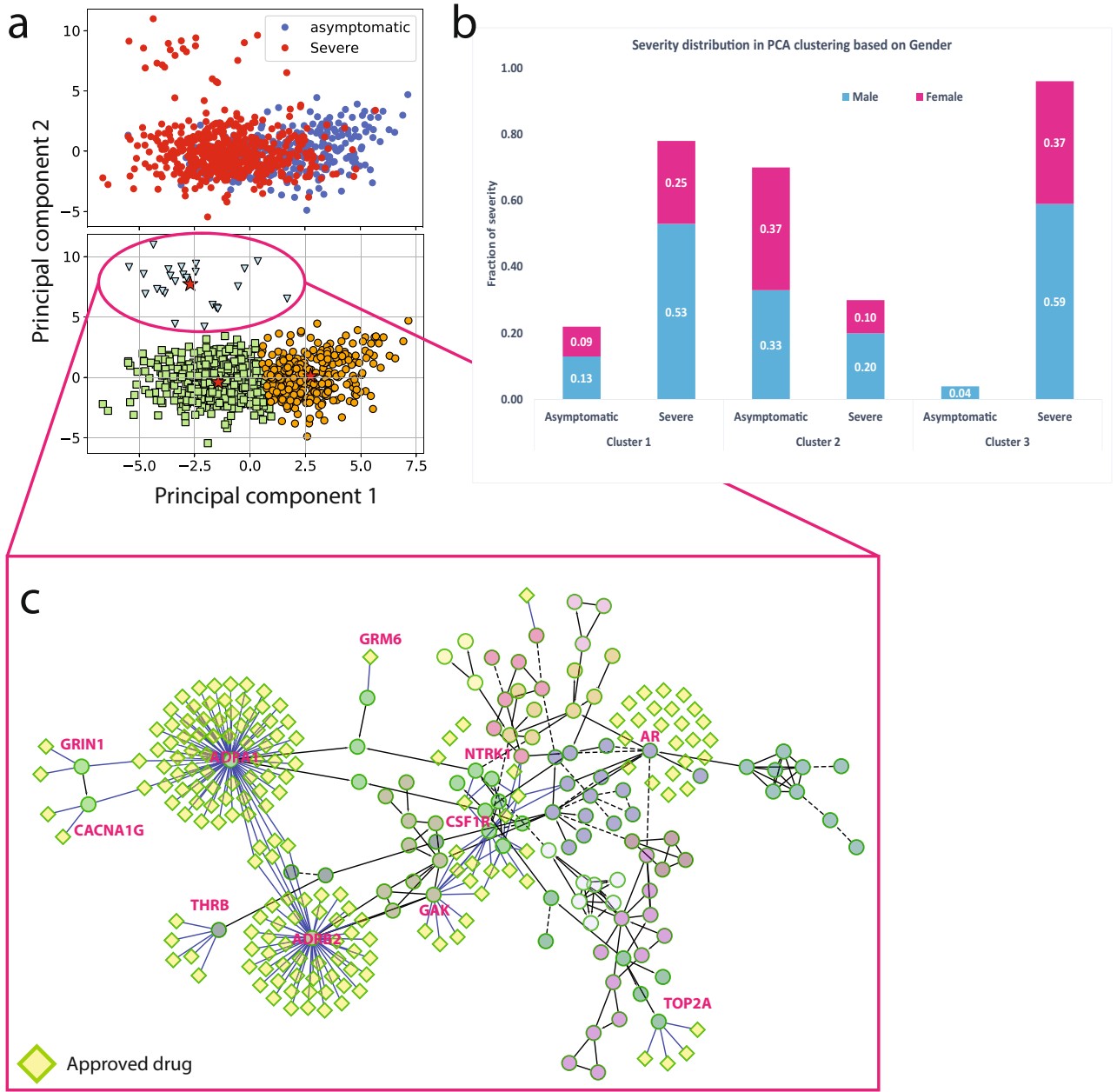

**Fig. 4 detection of distinct clinical groups via PCA and clustering. a** Projection of samples ($n = 841$) along the 1st and 2nd principal components and coloring based on severity (up) or clusters identified via k-means (bottom); **b** gender and clinical group composition of the clusters detected via k-means on the 1st and 2nd PCA components; **c** FI network constructed using mutated genes on the cluster of more severe patients and approved drugs available for any of these genes.

"*Abnormalities of breathing*" phenotype ($p$ val = 4E–06 and $p$ val = 1.6E–04, respectively), suggesting that patients carrying these variants might be at higher risk due to pre-existing difficulties of breathing (Fig. S6; Data S12).

Other general categories of traits that might be linked to severe COVID-19, such as "*Cardiovascular disease*" or "*Infectious disease*" showed similar distributions of associations of risk or mitigation factors (Fig. S7). Interestingly other categories, such as "*Integumentary system disease*" showed instead a prevalence of associations with mitigation factors (Fig. S7).

## Discussion
In this study, we have set up a multifaceted computational strategy to dissect patient genetic variants which might interplay

with the SARS-Cov-2 virus to increase the risk of, or to protect from, a severe response to infection.

We integrated into a stratified $k$-fold scheme a pipeline to perform variant features screening followed by machine learning model training and testing to robustly identify variants associated with severe response to COVID-19 infection. Our pipeline allowed a drastic reduction of the initial number of variants by several orders of magnitudes: from an initial set of approximately 1 M unique variants derived from WES to 1k variants receiving non-zero feature importance in at least one of the tree-based models. By only considering the variants with full support, i.e. always found to have non-zero feature importance in all the tree-based models, we further reduced the pool to only 16 variants. Models retrained with only full-support variants (plus age and gender as covariates) achieved superior performances (median

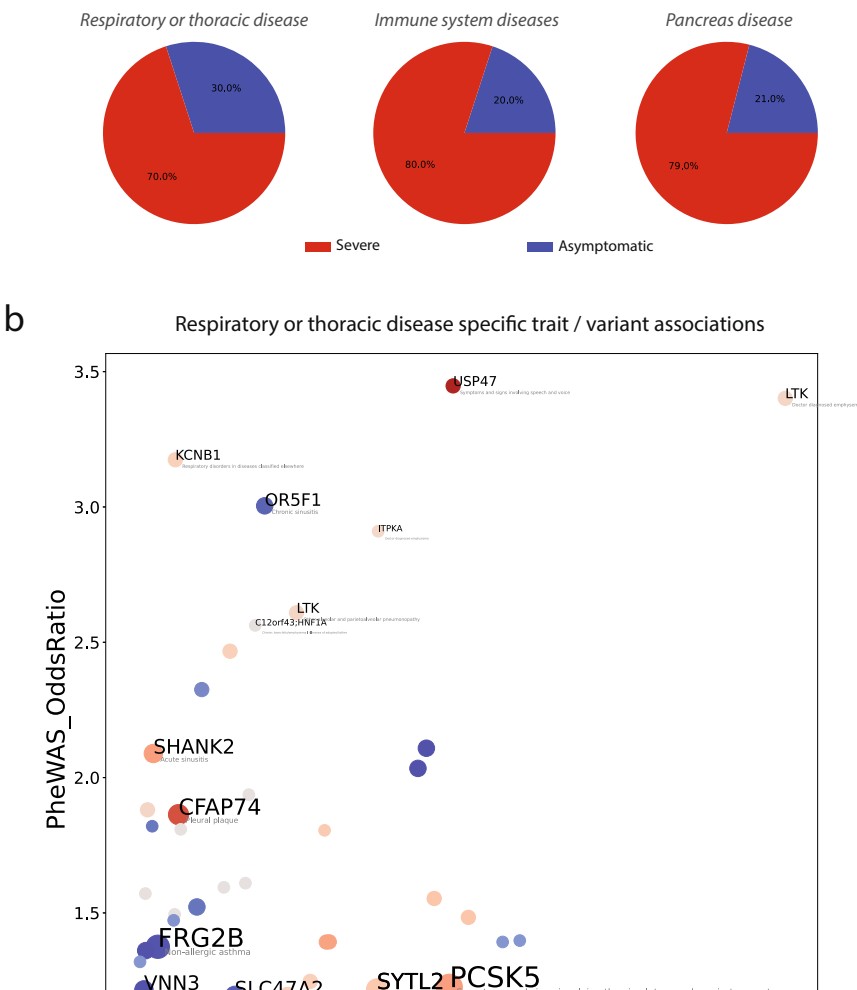

**Fig. 5 PheWAS analysis of most important variants. a** Phenotype categories displaying the greatest fraction of specific trait associations with variants enriched in severe versus asymptomatic patients; **b** scatter plot showing variant-specific traits associated within the "Respiratory or thoracic disease category". Dot diameter is proportional to the model support for each variant. The color is proportional to the log-odds ratio of the variant in the two groups of the cohort. Labels are printed only associations with PheWAS *P* value <0.001 and PheWAS oddsratio >2.5 or for variants having non-zero coefficients in at least one XGBoost model.

AUCROC = 86%, best model AUCROC = 91%). Although models trained with only patients age and gender already showed good performances in severity prediction (median AUCROC = 80%), confirming the predictive power of these covariates, the increase in performance followed by the inclusion of curated genetic information provides the foundation for integrated tools for COVID-19 severity forecast and patient stratification. When tested on a follow-up cohort of more than 600 our models achieved remarkable performances in identifying severe patients with good accuracy (ACC = 81.88% and AUCROC = 96%), performing considerably better than the ones obtained by training with only covariates or variants (Fig. 2e; Data S4).

The interpretability of our models allowed us to shed new light on the complex landscape of genetic interactions with virus genetics which contributed to a severe response to COVID-19 in an Italian cohort. Among the 16 variants with 100% support, only 6 genes (37%) were annotated in the largest

pathway knowledgebase, i.e. Reactome[22], suggesting that unannotated variants might modulate the interaction with the virus through yet-to-be-discovered biological mechanisms. Intriguingly, we found that two of these highly supported variants, i.e. chr9:34557898:A:T (*CNTFR*) and rs150021157 (*PCSK5*) interact within the second-largest module identified on the interaction network of the genes affected by mutations within our study. This cluster, which is moreover the only one characterized by two fully supported variants, is highly enriched in pathways linked to immune response and inflammation, such as the such as JAK-STAT signaling pathway, Cytokine-cytokine receptor interaction, and Interleukin-6 family signaling. The third cluster, which crosstalks with the second one, involves processes related to SMAD and TGF-β signaling, which were previously shown to be modulated by SARS nucleocapsid proteins[23].

We found that variants enriched in severe patients are involved in cardiomyopathies processes, supporting the established notion

that patients with heart disease or its risk factors are at greater risk of severe consequences following COVID-19 infection, including hospitalization, ventilation, or death[24]. Additional processes significantly enriched among severe mutations was ECM, whose importance in mediating the interaction with viral particles have been highlighted by affinity-purification proteomics experiments[25]. Recent experiments also confirmed a role for integrins in binding to UV-inactivated viral particles, through which outside-inside signaling is elicited via binding to $G\alpha13$[26]. Vesicle-mediated transport, such as clathrin-mediated endocytosis, has been shown to mediate a key entry point for SARS[27]. The latter pathway has also been confirmed to drive a chronic immune response in severe COVID-19[28]. Moreover, C-type leptin receptors have been shown to engage with the virus inducing robust pro-inflammatory responses in myeloid cells that correlated with COVID-19 severity[29].

On the other hand, some of the processes that we found significantly enriched among asymptomatic patients have been previously put in connection to SARS viral infection. For example, members of the machinery for DNA damage response have been shown to interact and affect the response to several DNA and RNA viruses[30] and it has been recently demonstrated that these pathways are also triggered by SARS-CoV-2 in vitro cellular models[31]. The Fanconi anemia pathway is tightly linked to DNA repair processes involving homologous recombination and genome integrity[32]. We therefore speculate that patients carrying variants on these pathways might differently interact with the virus, modulating a milder response to viral infection.

Several identified processes offer druggable options for therapeutic treatment. Androgen receptor signaling and its genetic variability have been already linked to COVID-19 severity[11,33] and its inhibition proposed as a therapeutic strategy (e.g[34].). We found several GPCR signaling instances significantly enriched in our network, in particular those related to $G_i$ and $G_q$ signaling, which mediate vascular inflammation. In particular, the $G_q$ pathway contributes to regulating calcium signaling, which is one of the most enriched processes in our dataset and which leads to endothelial barrier disruption via adherens junction disassembly[35]. On the other hand, $G_q$ signaling might also contribute to transactivate JAK-STAT pathway via (ERK)1/2 signaling[35], the latter in turn also activated by $G_i$ signaling[36]. It has also been recently shown that the C5a–C5aR1 axis, which also signals intracellularly through $G_q$, plays a key role in the pathophysiology of ARDS associated with COVID-19 by starting and maintaining several inflammatory responses through the recruitment and activation of neutrophils and monocytes[37]. Hence, similarly to what we and others previously described in cancer[38], genetic factors converging on modulating common GPCR downstream signaling pathways might also contribute to the onset of the inflammatory response related to COVID-19, at the same time offering new therapeutic intervention options for patients with severe forms of COVID-19. The recent finding that autoantibodies targeting GPCRs are associated with COVID-19 severity[39], further strengthens these receptors as therapeutic candidates.

We found multiple, recurrent disease traits associated with the variants identified. The variants rs150021157 and rs140300753, characterized by full support during supervised learning, also provide an example of associations to phenotypes that might play a role in COVID-19 severity, such as "Abnormalities of breathing phenotype". Some categories show a prevalence of associations with risk factors, such as "respiratory or thoracic disease", including specific traits such as chronic bronchitis, emphysema or COPD (the latter also found in the "infectious disease" category). Other categories enriched for associations with variants enriched in severe patients are "immune system disorders", including traits such as immunodeficiency with antibody defects, or "pancreas

disease", including several instances mainly associated to Type 2 diabetes, which is a known risk factor for severe COVID-19[40] and whose molecular connection to cytokine storm inflammatory response has now begun to emerge[17,41]. Taken together, these results further corroborate our analysis.

Our model is complementary to previous and ongoing efforts entailing machine learning techniques (i.e. LASSO logistic regression models) and a boolean representation of genetic variants to identify the most informative features associated to severity to compile an Integrated PolyGenic Score for COVID-19 severity predictions[17,18]. While we expect that some of the variants identified in this study might be specific for the Italian population, we believe that our approach could be readily trained on different cohorts to identify additional biomarkers for patient stratification in the clinics. Our capability to understand and forecast the genetic factors contributing to COVID-19 disease severity will certainly benefit from the availability of larger sequencing cohorts, the usage of more advanced methods for case-control associations in WES studies, new methodological advancement in the explainable AI field, as well as on our prior- or data-driven knowledge of biological mechanisms linking genetic variants to disease phenotypes.

## Methods

**Dataset and pre-processing**. We used the whole-exome sequencing (WES) dataset of 1982 European descent patients collected from the GEN-COVID Multicenter Study group coordinated by the University of Siena (https://clinicaltrials.gov/ct2/show/NCT04549831)[14]. Briefly, the GEN-COVID Multicenter Study includes a network of 22 Italian hospitals as well as local healthcare units and departments of preventative medicine (https://sites.google.com/dbm.unisi.it/gen-covid). It started its activity on March 16, 2020, following approval by the Ethical Review Board of the Promoter Center, University of Siena (Protocol n. 16917, approval dated March 16, 2020). Written informed consent was obtained from all individuals who contributed samples and data. Detailed clinical and laboratory characteristics (data), specifically related to COVID-19, were collected for all subjects.

Specifically, the WES dataset contained a total of 1.057 M unique simple variants. Patients were classified according to the grading scheme by the World Health Organization (WHO). The grading classification contained the following categories: 0=not hospitalized (a- or pauci-symptomatic); 1=hospitalized without respiratory support; 2=hospitalized O2 supplementation; 3=hospitalized CPAP-biPAP; 4= hospitalized intubated; 5=dead. We considered patients from more severe groups, i.e. 3,4, and 5, as cases, and asymptomatic patients from group 0, as controls, for a total of 1078 patients. We further refined the grading classification based on an ordinal logistic model which uses age as input feature for sex-stratified patients[17] and we retained only those patients whose grading classification was concordant with the one adjusted by age. This yielded a final set of 841 samples for downstream analysis.

**Statistics and reproducibility**. We employed the cohort of 841 patients to identify variants most associated to COVID-19 severity which we used, along with clinical co-variates such as age and sex, to train and test supervised binary classifiers of severity. We finally tested our ensemble of predictors on two unseen cohorts of patients: 618 individuals (122 asymptomatic, 496 severe), from a follow-up cohort of sequenced patients, and a set of 375 unique patients that were excluded from the original as well as the follow-up cohort due to inconsistencies between the original WHO grading classification and the one outputted by an ordinal logistic regression adjusted by age[17].

We detail below the statistical procedure employed.

**Stratified K-fold split of sample cohort into train and test sets**. We embedded a strategy for variant screening into a *stratified* five fold cross-validation scheme (using the *StratifiedKFold* function from the *scikit-learn* library https://scikit-learn.org/) to generate 5 random training and testing set splits of the original dataset. Each fold was constituted by a training set, corresponding to 80% of the dataset, which was also employed for variant screening and a remaining 20% for the testing set. The variants in the test set were curated from the variants screened in the training set. Through the stratified fivefold approach, we made sure that all the samples of the dataset were employed for testing.

**Variant screening**. GATK best-practices for germline variant calling pipeline were employed, as described in our previous work aimed at characterizing common, low-frequency, rare, and ultra-rare coding variants contribute to COVID-19 severity[17]. We employed a Log-Odds Ratio (LOR) statistics calculated on a 2×2

contingency Data to perform case-control association and to screen variants associated with either severe or asymptomatic patients in each of the training sets for each of the five folds generated. We grouped severe patients from clinical groups 5, 4, and 3 which were contrasted against the asymptomatic ones, considered as controls (group 0). We defined a contingency Data to measure the enrichment of reference (*Ref*) or alternative (*Alt*) alleles in either severe or control groups by employing an additive model, whereby homozygous genotype (1/1) has twice the risk (or protection) of the heterozygous type (0/1 or 1/0). We employed the *Data2x2* function from the *statsmodels* library (https://www.statsmodels.org/sData/index.html) to calculate LORs values and associated p-values and confidence intervals from the contingency Data in Fig. S8, respectively employing the functions *log_oddsratio*, *log_oddsratio_pvalue()* and *log_oddsratio_confint()*. We filtered variants with the following characteristics: $p-value < 0.05$ and $|LOR| \geq 1$. Variants with LOR > 1 are enriched among severe, while those with LOR < -1 are enriched among asymptomatics.

**Feature matrix generation**. For each split, we generated a feature matrix for the training set by assigning the allele counts of each screened variant for each sample of the training: i.e. 0 for genotype 0/0, 1 for genotypes 1/0 or 0/1, 2 for genotype 1/1. The feature matrix for the test set was defined by considering only variants identified as significant after screening the training set of the corresponding split and by assigning the allele count of each sample of the test set. We also included as additional features age, which was normalized, and gender, which was binarized by setting males to 0 and females to 1. Severe patients from group "$3 + 4 + 5$" were given the classification label "1", the asymptomatic patients from group 0 were given the label "0".

**Feature selection (removal of multicollinearity)**. We employed feature selection techniques to further reduce the number of considered features initially screened through the Log-Odds-Ratio statistics. We tried several approaches, including Lasso, ElasticNet and Multicollinearity, in combination with supervised training approaches (see below). After training several classifiers with the variants selected with each of these methods on a smaller cohort of 1200 samples, we found that removing multicollinearity from features by considering variant allele counts with correlation coefficients (corr.≤|0.8|) gave the best results. The screened features with little or no effects of multicollinearity formed the final 80% training sets in each fold and the final 20% corresponding validation sets used for training the supervised machine learning models.

**Supervised binary classification**. We trained supervised learning models for binary classification tasks by employing several algorithms, i.e. Support Vector Machine, Logistic Regression, Random Forest, and Extreme Gradient Boosting classifiers, available within the scikit-learn python library (https://scikit-learn.org/).

*Support Vector Classifier (SVC):* a popular machine learning method that classifies data points utilizing the concept of hyper-plan and kernel tricks to find fits that best separate the data cloud. In this study, we used the popular Jupyter notebook and *scikit-learn* python package to import the *"sklearn.svm"* SVC classifier model. We first set the SVC default regularization parameter "*C*" to 1, the class weight to "balanced" in order to account for imbalanced classification problems in the dataset. The default linear kernel was used first with the prediction probability set to true. The *GridSearchCV* was used to select the best hyperparameter values for the estimator "*C*", "*gamma*", and the kernel (Linear, Radial Basis Function (RBF), and polynomial) that are critical to the performance of the SVC classifier. The best *GridSearchCV* estimator hyperparameter values that were used to train our dataset were identified as the RBF kernel, $C = 10$, and *gamma* set to *0.1*.

*Logistic Regression:* a binary classification regression model that uses the logistic function to estimate the parameters of the logistic model. We import from the *scikit-learn* package the *"sklearn.linear_model"* the Logistic Regression model function. We first set the default logistic model classifier parameters; "*class weight = balanced*", $C = 0.3$ and *solver = sag*. The best *GirdSearchCV* estimator values used to train our dataset uses the regularization penalty of l1 (Lasso), $C = 0.7$, and *solver = saga*.

*Random Forest* (RF): an ensemble learning method that employs a bagging strategy. Multiple decision trees are trained using the same learning algorithm, and then predictions are aggregated from the individual decision tree. From the *"sklearn.ensemble"* library, we import the Random Forest Classifier function. The RF default model parameters use a class weight set to "balanced", maximum depth (*max_depth*) of the decision trees was set to 80, the number of features (*max_features*) was set to 2, minimum samples (*min_samples_leaf*) leaf of 3, minimum samples split (*min_samples_split*) of 10, and the number of trees (*n_estimators*) in the forest was set to 300. The *GridSearchCV* best model estimator parameters were "*bootstrap = True*", "*max_depth*" = 110, "*max_features*" = 2, "*min_samples_leaf*" = 5, "*min_samples_split*" = 10, and "*n_estimators*" = 100.

*Extreme Gradient Boosted Trees classifier (XGBoost):* an ensemble learning classifier family that utilizes boosting strategy to combine a set of weak learners and delivers improved prediction accuracy. We import from the XGBoost package *"xgboost"* library and xgboost function. We defined the data matrix (training feature set and classification label). We set the default XGBoost classifier model

parameters class weight to "balanced", learning objective to *"binary logistic"*. The best *GridSearchCV* estimator parameters values we used to train the dataset were "*learning_rate*" = 0.01, "*max_depth*" = 3, "*n_estimators*" = 140.

In summary, for each of the four ML models, we performed a parameter optimization through grid search (*GridSearchCV*), using the *accuracy_score* during grid search as a scoring method. We performed a fivefolds cross-validation, by splitting 80% for training and 20% for validation in each fold, repeated three times, using the *StratifiedKFold* function with $n\_splits = 5$ and $n\_repeats = 3$. We also set the class weight parameter to "*balanced*" in each of the ML algorithms employed. Both model training and hyperparameters optimization was done with a Python Jupyter notebook interactive web-based development environment using the scikit-learn and the xgboost packages. Model performances on the testing set were evaluated through the following metrics: Accuracy, F1, Precision, Recall, Matthew correlation coefficient (MCC), AUCROC.

A consensus voting approach was used to aggregate validation prediction probability scores of the four ML algorithms (SVC, Logistic Regression, Random Forest, and XGBoost classifiers) from each of the (20%) testing sets from each fold by considering the median of the probability distribution collected from the ensemble of models. The features (variants) that received non-zero weight during training of the supervised ML methods (Random Forest and XGBoost classifiers) in each fold were combined across the fivefold for further interpretation.

We performed a randomization test (i.e. Salzberg's test) to assess over-fitting (Salzberg, 1997), where we replace the original phenotypic labels of the training matrix with randomly assigned labels while preserving the ratio of the number of positive (severe) and negative (asymptomatic) patients (Data S13).

**Feature importance scores**. The feature importance assigns weight scores to individual features that interact to predict a particular event in the model. Feature importance for RandomForest and XGBoost models were calculated as the mean decrease in impurity for the feature using the feature importances function from *xgboost*. The feature importance (weights) scores assigned from these models' predictions were aggregated across the fivefolds to prioritize variants according to their consensus importance across folds for further downstream analysis. In particular we defined the model support (ms) of a given variant as the fraction of tree-based models assigning non-zero feature importance during the training of the model.

**Final testing on a follow-up cohort**. We tested the best performing models trained using most supported variants with and without covariates on a followup cohort of sequenced, Italian patients. An initial set of 838 samples corresponding to grading groups 0, 3, 4 and 5 were refined by applying the same ordered logistic regression classification *adjusted_by_age*, which yielded a final set of 618 individuals (122 asymptomatic, 496 severe). We generated an additional testing test by considering all the samples that were previously excluded due to inconsistency between the original WHO grading classification and the one outputted by an ordinal logistic regression adjusted by age classifier[17]. In details, in the original cohort that we used for training the model, there were 237 samples from either asymptomatic (grading 0) or severe (grading $3 + 4 + 5$) patients that were excluded due to classification inconsistencies, while in the follow-up cohort used for final testing of the model, 220 more individuals were excluded according to the same criteria. After removing patients with missing values, we obtained an aggregated list of 375 unique patients. We curated the allele counts of the 16 most informative variants, identified in the first stage of the analysis and model training, from this new set of patients and we used them, together with age and gender, as features for the testing. We evaluated the performances of the ensemble of the 20 models both on an individual as well as on an aggregated level, by calculating aggregated metrics obtained from the median of the probability distribution outputted by the ensemble of the 20 models on the testing samples.

**Principal component analysis (PCA) and clustering**. The variants with non-zero weights from best performing tree-based models were remapped back into the feature space to form a new feature count matrix covering 100% of the samples (i.e. 841 individuals). This reduced feature matrix was analyzed using Principal Component Analysis (PCA) techniques to reduce the dimensional space. In order for us to do this, we utilized the *"sklearn.decomposition"* library to import the PCA function. We standardized the feature count matrix using the *"sklearn.preprocessing"* library to import the Standard Scaler function. We transformed the normal feature count matrix considering the 1st and 2nd PCA components. We further employ the K-means clustering technique (using the *"sklearn.cluster"* library to import the *"KMeans"* function) to visualize and cluster the 2D PCA components (1st and 2nd dimensions). We set the default cluster size to 3, the maximum iteration (max_iter=1000), and a tolerance value (tol=1E–04). Clusters of patients that express interesting severity patterns were further analyzed using the pathway enrichment for biological interpretations and implications.

**Pathway enrichment analysis**. The pathway enrichment analysis was done using the ReactomeFIViz plugin[42] available in Cytoscape[43]. The genes corresponding to variants with non-zero feature importance from XGBoost were used to construct a Functional Interaction (FI) network. The general FI network comprised all the genes affected by variants with non-zero feature importances in both patient

groups. Node diameter is proportional to the number of variants with non-zero coefficients in any tree-based models. Node color is instead proportional to the LOR with the highest absolute value among the variants associated with a given gene. Modules within the network were identified through spectral partition clustering[44]. Reactome pathways over-representation analysis (FDR<0.1) was calculated on either the whole network or for each individual module. We also generated group-specific networks by keeping separated genes with variants enriched in severity from those enriched in asymptomatic and performed pathway over-representation analysis (FDR < 0.1) on the distinct networks.

**Retrieving associations between variants and disease traits or phenotypes**. We retrieved associations among the variants identified in our study and disease traits or phenotypes through the Open Targets Genetics platform[21]. We interrogated the database using the GraphQL query language embedded in a python script and by inputting the variant coordinates (given by chromosome nr, position, Ref, and Alt allele). For each PheWAS association, we retrieved the following data: *eaf, beta, se, nTotal, nCases, oddsRatio, studyId* and *pval*. Only PheWAS with *oddsRatio* > 1 and *p* val <0.001 were considered. The statistics were done only for the variants with non-zero feature importance from XGBoost models.

All the analyses were performed using customized Python (v3.8) scripts, with the following libraries: *scipy (v1.2.0), numpy (v1.19.4), scikit-learn (v0.23.2.), statsmodels (v0.11.0)* and *matplotlib (v3.2.1)*.

**Reporting summary**. Further information on research design is available in the Nature Research Reporting Summary linked to this article.

## Data availability

All the data and scripts to generate the figures are available, in a dedicated folder for each figure, at the following URL: https://github.com/raimondilab/An-explainable-model-of-host-genetic-interactions-linked-to-Covid19-severity/tree/main/scripts_figures_manuscript_COVID_19. The source data for graph and charts are provided in Supplementary Data 1–13.

## Code availability

All the scripts and models generated and data to reproduce them are available at the following URL: https://github.com/raimondilab/An-explainable-model-of-host-genetic-interactions-linked-to-Covid19-severity

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

## Acknowledgements

We are grateful to Prof. Luigi Ambrosio for his initiative and for helpful discussions. F.R. was supported by the Italian Ministry of University and Research through the Department of excellence "Faculty of Sciences" of Scuola Normale Superiore. We gratefully acknowledge computational resources of the Center for High Performance Computing (CHPC) at SNS. This study is part of the GEN-COVID Multicenter Study, https://sites.google.com/dbm.unisi.it/gen-COVID, the Italian multicenter study aimed at identifying the COVID-19 host genetic bases. Specimens were provided by the COVID-19 Biobank of Siena, which is part of the Genetic Biobank of Siena, member of BBMRI-IT, of Telethon Network of Genetic Biobanks (project no. GTB18001), of EuroBioBank, and of RD-Connect. We thank the CINECA consortium for providing computational resources and the Network for Italian Genomes (NIG; http://www.nig.cineca.it) for its support. We thank private donors for the support provided to AR (Department of Medical Biotechnologies, University of Siena) for the COVID-19 host genetics research project (D.L n.18 of March 17, 2020). We also thank the COVID-19 Host Genetics Initiative (https://www.COVID-19hg.org/), MIUR project 'Dipartimenti di Eccellenza 2018–2020' to the Department of Medical Biotechnologies University of Siena, Italy, and 'Bando Ricerca COVID-19 Toscana' project to Azienda Ospedaliero-Universitaria Senese. We thank Intesa San Paolo for the 2020 charity fund dedicated to the project N B/2020/0119 'Identificazione delle basi genetiche determinanti la variabilità clinica della risposta a COVID-19 nella popolazione italiana'. We thank EU project H2020-SC1-FA-DTS-2018-2020, entitled "International consortium for integrative genomics prediction (INTERVENE)" - Grant Agreement No. 101016775.

## Author contributions

Performed data analysis: A.O., F.R. Designed the experiments: F.Chiaromonte, A.R., S.F., F.R. Provided biological samples and clinical data: M.B., F.F., GEN-COVID Multicenter Study. Wrote or contributed to the writing of the manuscript: A.O., C.Fellarini, F. Colombo, F.F., N.P., M.B., S.F., F.Chiaromonte, F.R.

## Competing interests

The authors declare no competing interests.

## Additional information

## GEN-COVID Multicenter Study

Francesca Mari[4,5,6], Sergio Daga[4,5], Elisa Benetti[4], Mirella Bruttini[4,5,6], Maria Palmieri[4,5], Susanna Croci[4,5], Sara Amitrano[6], Ilaria Meloni[4,5], Elisa Frullanti[4,5], Gabriella Doddato[4,5], Mirjam Lista[4,5], Giada Beligni[4,5], Floriana Valentino[4,5], Kristina Zguro[4], Rossella Tita[6], Annarita Giliberti[4,5], Maria Antonietta Mencarelli[6], Caterina Lo Rizzo[6], Anna Maria Pinto[6], Francesca Ariani[4,5,6], Laura Di Sarno[4,5], Francesca Montagnani[4,10], Mario Tumbarello[4,10], Ilaria Rancan[4,10], Massimiliano Fabbiani[10], Barbara Rossetti[10], Laura Bergantini[11], Miriana D'Alessandro[11], Paolo Cameli[11], David Bennett[11], Federico Anedda[12], Simona Marcantonio[12], Sabino Scolletta[12], Federico Franchi[12], Maria Antonietta Mazzei[13], Susanna Guerrini[13], Edoardo Conticini[14], Luca Cantarini[14], Bruno Frediani[14], Danilo Tacconi[15], Chiara Spertilli Raffaelli[15], Marco Feri[16], Alice Donati[16], Raffaele Scala[17], Luca Guidelli[17], Genni Spargi[18], Marta Corridi[18], Cesira Nencioni[19], Leonardo Croci[19], Gian Piero Caldarelli[20], Davide Romani[21], Paolo Piacentini[21], Maria Bandini[21], Elena Desanctis[21], Silvia Cappelli[21], Anna Canaccini[22], Agnese Verzuri[22], Valentina Anemoli[22], Manola Pisani[22], Agostino Ognibene[23], Alessandro Pancrazzi[23], Maria Lorubbio[23], Massimo Vaghi[24], Antonella D'Arminio Monforte[25], Federica Gaia Miraglia[25], Raffaele Bruno[26,27], Marco Vecchia[26], Massimo Girardis[28], Sophie Venturelli[28], Stefano Busani[28], Andrea Cossarizza[29], Andrea Antinori[30], Alessandra Vergori[30], Arianna Emiliozzi[30], Stefano Rusconi[31,32], Matteo Siano[32], Arianna Gabrieli[32], Agostino Riva[31,32], Daniela Francisci[33], Elisabetta Schiaroli[33], Francesco Paciosi[33], Andrea Tommasi[33], Umberto Zuccon[34], Lucia Vietri[34], Pier Giorgio Scotton[35], Francesca Andretta[35], Sandro Panese[36], Stefano Baratti[36], Renzo Scaggiante[37], Francesca Gatti[37], Saverio Giuseppe Parisi[38], Francesco Castelli[39], Eugenia Quiros-Roldan[39],

Melania Degli Antoni[39], Isabella Zanella[40,41], Matteo Della Monica[42], Carmelo Piscopo[42], Mario Capasso[43,44,45], Roberta Russo[43,44], Immacolata Andolfo[43,44], Achille Iolascon[43,44], Giuseppe Fiorentino[46], Massimo Carella[47], Marco Castori[47], Filippo Aucella[48], Pamela Raggi[49], Rita Perna[49], Matteo Bassetti[50,51], Antonio Di Biagio[50,51], Maurizio Sanguinetti[52,53], Luca Masucci[52,53], Alessandra Guarnaccia[52], Serafina Valente[54], Oreste De Vivo[54], Elena Bargagli[11], Marco Mandalà[55], Alessia Giorli[55], Lorenzo Salerni[55], Patrizia Zucchi[56], Pierpaolo Parravicini[56], Elisabetta Menatti[57], Tullio Trotta[58], Ferdinando Giannattasio[58], Gabriella Coiro[58], Fabio Lena[59], Gianluca Lacerenza[59], Domenico A. Coviello[60], Cristina Mussini[61], Enrico Martinelli[62], Luisa Tavecchia[63], Mary Ann Belli[63], Lia Crotti[64,65,66,67,68], Gianfranco Parati[64,65], Maurizio Sanarico[69], Filippo Biscarini[70], Alessandra Stella[70], Marco Rizzi[71], Franco Maggiolo[71], Diego Ripamonti[71], Claudia Suardi[72], Tiziana Bachetti[73], Maria Teresa La Rovere[74], Simona Sarzi-Braga[75], Maurizio Bussotti[76], Katia Capitani[4,77], Simona Dei[78], Sabrina Ravaglia[79], Rosangela Artuso[80], Elena Andreucci[80], Giulia Gori[80], Angelica Pagliazzi[80], Erika Fiorentini[80], Antonio Perrella[81], Francesco Bianchi[81,4], Paola Bergomi[82], Emanuele Catena[82], Riccardo Colombo[82], Sauro Luchi[83], Giovanna Morelli[83], Paola Petrocelli[83], Sarah Iacopini[83], Sara Modica[83], Silvia Baroni[84], Francesco Vladimiro Segala[85], Francesco Menichetti[86], Marco Falcone[86], Giusy Tiseo[86], Chiara Barbieri[86], Tommaso Matucci[86], Davide Grassi[87], Claudio Ferri[87], Franco Marinangeli[88], Francesco Brancati[89], Antonella Vincenti[90], Valentina Borgo[90], Stefania Lombardi[90], Mirco Lenzi[90], Massimo Antonio Di Pietro[91], Francesca Vichi[91], Benedetta Romanin[91], Letizia Attala[91], Cecilia Costa[91], Andrea Gabbuti[91], Roberto Menè[64,65], Marta Colaneri[26], Patrizia Casprini[92], Giuseppe Merla[93,94], Gabriella Maria Squeo[93], Marcello Maffezzoni[95], Stefania Mantovani[96], Mario U. Mondelli[96] & Serena Ludovisi[97]

[10]Department of Medical Sciences, Infectious and Tropical Diseases Unit, Azienda Ospedaliera Universitaria Senese, Siena, Italy. [11]Unit of Respiratory Diseases and Lung Transplantation, Department of Internal and Specialist Medicine, University of Siena, Siena, Italy. [12]Department of Emergency and Urgency, Medicine, Surgery and Neurosciences, Unit of Intensive Care Medicine, Siena University Hospital, Siena, Italy. [13]Department of Medical, Surgical and Neuro Sciences and Radiological Sciences, Unit of Diagnostic Imaging, University of Siena, Siena, Italy. [14]Rheumatology Unit, Department of Medicine, Surgery and Neurosciences, University of Siena, Policlinico Le Scotte, Siena, Italy. [15]Department of Specialized and Internal Medicine, Infectious Diseases Unit, San Donato Hospital Arezzo, Arezzo, Italy. [16]Department of Emergency, Anesthesia Unit, San Donato Hospital, Arezzo, Italy. [17]Department of Specialized and Internal Medicine, Pneumology Unit and UTIP, San Donato Hospital, Arezzo, Italy. [18]Department of Emergency, Anesthesia Unit, Misericordia Hospital, Grosseto, Italy. [19]Department of Specialized and Internal Medicine, Infectious Diseases Unit, Misericordia Hospital, Grosseto, Italy. [20]Clinical Chemical Analysis Laboratory, Misericordia Hospital, Grosseto, Italy. [21]Dipartimento di Prevenzione, Azienda USL Toscana Sud Est, Tuscany, Italy. [22]Dipartimento Tecnico-Scientifico Territoriale, Azienda USL Toscana Sud Est, Tuscany, Italy. [23]Clinical Chemical Analysis Laboratory, San Donato Hospital, Arezzo, Italy. [24]Chirurgia Vascolare, Ospedale Maggiore di Crema, Crema, Italy. [25]Department of Health Sciences, Clinic of Infectious Diseases, ASST Santi Paolo e Carlo, University of Milan, Milano, Italy. [26]Division of Infectious Diseases I, Fondazione IRCCS Policlinico San Matteo, Pavia, Italy. [27]Department of Clinical, Surgical, Diagnostic, and Pediatric Sciences, University of Pavia, Pavia, Italy. [28]Department of Anesthesia and Intensive Care, University of Modena and Reggio Emilia, Modena, Italy. [29]Department of Medical and Surgical Sciences for Children and Adults, University of Modena and Reggio Emilia, Modena, Italy. [30]HIV/AIDS Department, National Institute for Infectious Diseases, IRCCS, Lazzaro Spallanzani, Rome, Italy. [31]III Infectious Diseases Unit, ASST-FBF-Sacco, Milan, Italy. [32]Department of Biomedical and Clinical Sciences Luigi Sacco, University of Milan, Milan, Italy. [33]Infectious Diseases Clinic, "Santa Maria" Hospital, University of Perugia, Perugia, Italy. [34]Respiratory Diseases Unit, "Santa Maria degli Angeli" Hospital, Pordenone, Italy. [35]Department of Infectious Diseases, Treviso Hospital, Local Health Unit 2 Marca Trevigiana, Treviso, Italy. [36]Clinical Infectious Diseases, Mestre Hospital, Venezia, Italy. [37]Infectious Diseases Clinic, ULSS1 Belluno, Italy. [38]Department of Molecular Medicine, University of Padova, Padova, Italy. [39]Department of Infectious and Tropical Diseases, University of Brescia and ASST Spedali Civili Hospital, Brescia, Italy. [40]Department of Molecular and Translational Medicine, University of Brescia, Brescia, Italy. [41]Clinical Chemistry Laboratory, Cytogenetics and Molecular Genetics Section, Diagnostic Department, ASST Spedali Civili di Brescia, Brescia, Italy. [42]Medical Genetics and Laboratory of Medical Genetics Unit, A.O.R.N. "Antonio Cardarelli", Naples, Italy. [43]Department of Molecular Medicine and Medical Biotechnology, University of Naples Federico II, Naples, Italy. [44]CEINGE Biotecnologie Avanzate, Naples, Italy. [45]IRCCS SDN, Naples, Italy. [46]Unit of Respiratory Physiopathology, AORN dei Colli, Monaldi Hospital, Naples, Italy. [47]Division of Medical Genetics, Fondazione IRCCS Casa Sollievo della Sofferenza Hospital, San Giovanni Rotondo, Italy. [48]Department of Medical Sciences, Fondazione IRCCS Casa Sollievo della Sofferenza Hospital, San Giovanni Rotondo, Italy. [49]Clinical Trial Office, Fondazione IRCCS Casa Sollievo della Sofferenza Hospital, San Giovanni Rotondo, Italy. [50]Department of Health Sciences, University of Genova, Genova, Italy. [51]Infectious Diseases Clinic, Policlinico San Martino Hospital, IRCCS for Cancer Research Genova, Genova, Italy. [52]Microbiology, Fondazione Policlinico Universitario Agostino Gemelli IRCCS, Catholic University of Medicine, Rome, Italy. [53]Department of Laboratory Sciences and Infectious Diseases, Fondazione Policlinico Universitario A. Gemelli IRCCS, Rome, Italy. [54]Department of Cardiovascular Diseases, University of Siena, Siena, Italy. [55]Otolaryngology Unit, University of Siena, Siena, Italy. [56]Department of Internal Medicine, ASST Valtellina e Alto Lario, Sondrio, Italy. [57]Study Coordinator Oncologia Medica e Ufficio Flussi, Sondrio, Italy. [58]First Aid Department, Luigi Curto Hospital, Polla, Salerno, Italy. [59]Department of Pharmaceutical Medicine, Misericordia Hospital, Grosseto, Italy. [60]U.O.C. Laboratorio di Genetica

Umana, IRCCS Istituto G. Gaslini, Genova, Italy. [61]Infectious Diseases Clinics, University of Modena and Reggio Emilia, Modena, Italy. [62]Department of Respiratory Diseases, Azienda Ospedaliera di Cremona, Cremona, Italy. [63]U.O.C. Medicina, ASST Nord Milano, Ospedale Bassini, Cinisello Balsamo, MI, Italy. [64]Istituto Auxologico Italiano, IRCCS, Department of Cardiovascular, Neural and Metabolic Sciences, San Luca Hospital, Milan, Italy. [65]Department of Medicine and Surgery, University of Milano-Bicocca, Milan, Italy. [66]Istituto Auxologico Italiano, IRCCS, Center for Cardiac Arrhythmias of Genetic Origin, Milan, Italy. [67]Istituto Auxologico Italiano, IRCCS, Laboratory of Cardiovascular Genetics, Milan, Italy. [68]Member of the European Reference Network for Rare, Low Prevalence and Complex Diseases of the Heart-ERN GUARD-Heart, Milan, Italy. [69]Independent Data Scientist, Milan, Italy. [70]CNR-Consiglio Nazionale delle Ricerche, Istituto di Biologia e Biotecnologia Agraria (IBBA), Milano, Italy. [71]Unit of Infectious Diseases, ASST Papa Giovanni XXIII Hospital, Bergamo, Italy. [72]Fondazione per la ricerca Ospedale di Bergamo, Bergamo, Italy. [73]Direzione Scientifica, Istituti Clinici Scientifici Maugeri IRCCS, Pavia, Italy. [74]Istituti Clinici Scientifici Maugeri IRCCS, Department of Cardiology, Institute of Montescano, Pavia, Italy. [75]Istituti Clinici Scientifici Maugeri, IRCCS, Department of Cardiac Rehabilitation, Institute of Tradate, Tradate, VA, Italy. [76]Istituti Clinici Scientifici Maugeri IRCCS, Department of Cardiology, Institute of Milan, Milan, Italy. [77]Core Research Laboratory, ISPRO, Florence, Italy. [78]Health Management, Azienda USL Toscana Sudest, Tuscany, Italy. [79]IRCCS C. Mondino Foundation, Pavia, Italy. [80]Medical Genetics Unit, Meyer Children's University Hospital, Florence, Italy. [81]Department of Medicine, Pneumology Unit, Misericordia Hospital, Grosseto, Italy. [82]Department of Anesthesia and Intensive Care Unit, ASST Fatebenefratelli Sacco, Luigi Sacco Hospital, Polo Universitario, University of Milan, Milan, Italy. [83]Infectious Disease Unit, Hospital of Lucca, Lucca, Italy. [84]Department of Diagnostic and Laboratory Medicine, Institute of Biochemistry and Clinical Biochemistry, Fondazione Policlinico Universitario A. Gemelli IRCCS, Catholic University of the Sacred Heart, Rome, Italy. [85]Clinic of Infectious Diseases, Catholic University of the Sacred Heart, Rome, Italy. [86]Department of Clinical and Experimental Medicine, Infectious Diseases Unit, University of Pisa, Pisa, Italy. [87]Department of Clinical Medicine, Public Health, Life and Environment Sciences, University of L'Aquila, L'Aquila, Italy. [88]Anesthesiology and Intensive Care, University of L'Aquila, L'Aquila, Italy. [89]Medical Genetics Unit, Department of Life, Health and Environmental Sciences, University of L'Aquila, L'Aquila, Italy. [90]Infectious Disease Unit, Hospital of Massa, Massa, Italy. [91]Infectious Diseases Unit, Santa Maria Annunziata Hospital, USL Centro, Florence, Italy. [92]Laboratory of Clinical Pathology and Immunoallergy, Florence-Prato, Italy. [93]Laboratory of Regulatory and Functional Genomics, Fondazione IRCCS Casa Sollievo della Sofferenza, San Giovanni Rotondo, (Foggia), Italy. [94]Department of Molecular Medicine and Medical Biotechnology, University of Naples Federico II, Naples, Italy. [95]University of Pavia, Pavia, Italy. [96]Division of Clinical Immunology and Infectious Diseases, Department of Medicine, Fondazione IRCCS Policlinico San Matteo, Pavia, Italy. [97]Fondazione IRCCS Ca' Granda Ospedale Maggiore Policlinico, Milan, Italy.

