## [Peer Review File · Communications Biology]

Reviewers' comments:

Reviewer #1 (Remarks to the Author):

Overview:

This manuscript describes a whole exome sequencing (WES) study looking at ~2,000 patients with varying levels of COVID-19 severity in order to identify rare variants that may correlate with severity of disease. The analytic strategy employed by the investigators used multiple search strategies including machine learning approaches like feature importance analysis to identify and rank variants most associated with severity of disease, and the to predict severity based on these variants. The top 16 variants identified by all modeling approaches showed evidence of being involved in severity of the phenotype, including through follow-up PheWAS analyses of the variants which indicated that these same variants were also involved in respiratory diseases and disorders, suggesting biological plausibility for their role in severity of COVID pulmonary symptoms. Predictive modeling of these same genetic variants found only a modest improvement in prediction over the predictive ability of a model including just age and gender, AUCROC=0.86 (max AUCROC=0.91) vs. AUCROC=0.80, suggesting that genetic variants are useful in predictive modeling, but not as important as clinical covariates of importance. While this does provide an example of the utility of feature importance analysis, it may overstate the utility of identifying predictive genetic variants in the context of a disease like COVID for which much of the variability in phenotype expression is driven by comorbidities.

Overall, the concerns with the design of the study are relatively minor, however the linguistic/stylistic issues observed need to be fixed, as well as several key aspects of the Results section: (1) methods are heavily recapitulated in the Results section, whereas only key aspects of methods should be mentioned in Results and the most important methods information focused on in the Methods section; (2) the Results section is harder to follow because of the high-level of detail dedicated to each pathway reported, as well as support information on each pathway provided in Results. Additional evidence supporting the pathways identified in study that is not directly part of the work performed in the study should be reported in the Discussion section. Finally, while it is understandable that scientific writing in a non-native language is immensely challenging, and the authors have made a genuine effort in their writing to produce a readable article, it would be helpful to have a native or highly-fluent English speaker fix some of the grammatical issues identified as well as help with better word choice where phrasing is ambiguous and leads to potential misunderstandings; this would greatly improve the manuscript with relatively little effort. I have made an attempt to identify grammatical issues or sections that may potentially be misunderstood by readers.

Based on these considerations, it is recommended that the manuscript undergo substantial revision and resubmission for a second evaluation. The content is of sufficient interest to the genetics, feature analysis, and COVID research communities that a revised version of this manuscript would be a valuable addition to the COVID genetics literature.

Major Comments:

- 1) Formatting: It should be noted that the Nature Communications Biology reference notation is to use superscripted numerals (e.g., "1¹"), whereas the entirety of the text uses citations formatted in a first author-date format (e.g., "Pairo-Castineira et al., 2020"). This will need to be changed throughout the text.
- 2) Results: Throughout the Results section, methods are described in great detail rather than focusing exclusively on the salient findings of these analyses. For instance, the cross-validation approach employed is described over several sentences, but the model performance is described only in one sentence. This section should be revised to indicate clearly what the different models found, and the methodological descriptions folded into the Methods section. Only the most important aspects of modeling that are needed for the findings to be understood should be described in the Results section.

3) Results: This section lists the pathway findings and much background detail, but the abundance of information presented makes it hard to identify the findings of greatest importance. Where tables and figures can provide adequate information on the pathways with most significant associations, it may be useful to shorten the description of these findings and exclude descriptions of background details on all pathways or save those details for the Discussion section. This would tighten up the section and make findings more salient for the readers.

4) Methods: the software packages used are listed, but not cited, and it is unclear where certain packages (e.g., "Table2x2") are implemented, whether in R, scikit-learning, or elsewhere. These details are in the last sentence of methods, but might be clearer if they are reported earlier in the Methods section.

Minor Comments

1) Abstract (page 2, line 39): It is recommended to change "allowed to identify a handful of 16 variants" to "allowed us to identify 16 variants" or "helped us identify 16 variants". The expression 'handful of' does not typically include an actual number, and usually denotes fewer individual items than 10.

2) Abstract (page 2, line 42): It is recommended to change "good accuracy" to "high accuracy", as 'good' represents a qualitative description rather than 'high' which captures the level of or degree of something.

3) Abstract (page 2, line 42): It is recommended to replace "ROC_AUC" with "AUCROC" here, and wherever else is needed in the text.

4) Abstract (page 2, line 42): It is recommended to change "confirming their link with COVID-19 severity outcome" to "supporting their link with COVID-19 severity outcome" or even "strongly supporting". This is suggested because 'confirming' may be perceived as an indication of direct causality, and while there is strong evidence, additional phenotypic modeling is needed to confirm the function of these variants.

5) Introduction (page 3, lines 64-65): It is recommended to change "challenging at an unprecedented level health, economical and societal systems worldwide" to "challenging health, economic, and social systems worldwide at an unprecedented level."

6) Introduction (page 3, lines 68): Change "ARSD" to "ARDS" in all instances it is used.

7) Introduction (page 3, lines 75): A citation labeled "1" is missing- please add this in.

8) Introduction (page 3, lines 76-79): This is a run on-sentence. It is recommended to consider breaking it in two sentences, where the phrase "...also considered that" would start the new sentence, rewritten as something like "It is important to consider that,..." or "It should also be remembered that,..."

9) Introduction (page 3, line 81): It is recommended to change "genes involved in type I interferon (IFN)" to "genes involved in type I interferon (IFN) responses."

10) Introduction (page 3, line 83): Remove the word "campaigns." Alternatively, replace it with "projects."

11) Introduction (page 3, lines 85-86): Replace "Up to now,..." with "Until now,..."

12) Introduction (page 4, line 97): Change "While GWAS studies provide solid foundations..." to "While GWAS provide solid evidence..."

13) Introduction (page 4, lines 112-113): Change "to identify a few dozens of genetic variants" to "to identify a few dozen genetic variants".

14) Results (page 4, line 117): The Results section indicates that a WES dataset of "germline variants" was used, but according to the prior GEN-COVID paper, somatic cells were used for DNA. Please clarify where it is indicated that germline sequencing was performed.

15) Results (page 5, lines 160-161): Change the "F1-score" to "F1" here and "F1" or "F1 score" throughout the text.

16) Results (page 6, lines 179-182): Rephrase this sentence to flow more smoothly. For example, "We observed a high level of performance when we tested the ensemble of models trained with only informative variants..." The phrase "achieved good performances" is fine, but it is a qualitative description rather than a level of performance. I would also suggest removing "Remarkably" as it could be seen as editorializing, however this word could stay as it draws attention to the finding.

17) Results (page 6, lines 198-199): Change "associated to cardiomyopathies" to "associated with cardiomyopathies."

18) Results (throughout): Scientific notation for decimals are incorrectly reported and should be modified throughout. For instance, the FDR reported for the association of arrhythmogenic right ventricular cardiomyopathy is reported as "FDR=4.03e-05". In various program outputs the capital letter "E" stands for "Exponential Notation" and the number following it is not superscripted. Using a lowercase "e" and superscripted number indicates raising Euler's number, the mathematical constant approximately equal to 2.71828, raised to the power shown. So the value "FDR=4.03e-05" has the value FDR=0.027154 rather than the value the investigators are trying to indicate, which is FDR=0.0000403. Throughout the text, wherever it says "e-x", this should be changed to either "E-X" or " $\times 10^{-x}$ " (e.g., report "FDR=4.03e-05" instead as "FDR=4.03 $\times 10^{-5}$ ").

19) Results (page 7, line 211): Change "remarking" to "marking".

20) Results (page 7, line 222): Instead of reporting log Odds Ratios in the text, it is recommended to report them as Odds Ratios (OR) as those are easier to understand. Other than the log-odds ratio, these could be reported as their mathematical equivalent, β s. For instance, a $\log(\text{OR})=-1.34$ is easier to understand if described as an OR=0.26, which indicates that the association is highly protective.

21) Results (page 7, line 230): Change "All these three pathways are participated by" to "All three pathways include".

22) Results (page 7, line 233): Change "top20 of genes" to "top 20 genes".

23) Results (page 7, lines 236-238): It is unclear what is meant by "rs150021157... affecting the PCSK5 gene" if that means that "rs150021157... located in the PCSK5 gene". This may be a misreading on my part.

24) Results (page 8, line 266): The values "pval=0.002881778" should be reduced to three significant figures ("pval=0.00288") in keeping with results reported elsewhere in the paper. All other p-values should be set at three significant figures.

25) Results (page 8, line 268): Change "which entails CEP131" to "which includes CEP131". It is unclear whether the statement means that "affected by the variant rs2659015" means that the variant

is located in the gene or the variant affects that gene in some way.

26) Results (page 9, line 295): Change "three groups of patients on the original cohort" to "three groups of patients in the original cohort".

27) Results (page 9, line 298): Change "(70%) of the total" to "(70% of the total)".

28) Results (page 10, line 313): Change "associated to disease traits" to "associated with disease traits".

29) Results (page 10, line 344): It is unclear what "invariably relevant for severity classification" means- a different word choice may be helpful.

30) Discussion (page 11, line 379): Change "genetic interactions interplaying with virus genetics" to "genetic interactions with virus genetics".

31) Methods (page 14, line 486): Change "We employed a Log-Odds Ratio (LOR) statistics to perform case-control association" to something like "We used logistic regression to estimate log-Odds Ratios (LORs) for testing association in our case-control datasets". Without this change, it is less clear how the analysis was done.

Reviewer #2 (Remarks to the Author):

The paper claim to have identified variants with nonlinear interactions associated with severe covid. The main difficulty with this paper is the method of patient selection that uses age and sex to predict the outcome. This will make the analysis prone to cohort effects. Therefore, the differences in variant frequency between cases and controls could be attributed to differences in allele frequencies between older and younger patients. Hence, all analyses need to be repeated without making the(recursive) selection based on the same sample used to identify variants.

Reviewers' comments:

Reviewer #1 (Remarks to the Author):

Overview:

This manuscript describes a whole exome sequencing (WES) study looking at ~2,000 patients with varying levels of COVID-19 severity in order to identify rare variants that may correlate with severity of disease. The analytic strategy employed by the investigators used multiple search strategies including machine learning approaches like feature importance analysis to identify and rank variants most associated with severity of disease, and the to predict severity based on these variants. The top 16 variants identified by all modeling approaches showed evidence of being involved in severity of the phenotype, including through follow-up PheWAS analyses of the variants which indicated that these same variants were also involved in respiratory diseases and disorders, suggesting biological plausibility for their role in severity of COVID pulmonary symptoms. Predictive modeling of these same genetic variants found only a modest improvement in prediction over the predictive ability of a model including just age and gender, AUCROC=0.86 (max AUCROC=0.91) vs. AUCROC=0.80, suggesting that genetic variants are useful in predictive modeling, but not as important as clinical covariates of importance. While this does provide an example of the utility of feature importance analysis, it may overstate the utility of identifying predictive genetic variants in the context of a disease like COVID for which much of the variability in phenotype expression is driven by comorbidities.

Overall, the concerns with the design of the study are relatively minor, however the linguistic/stylistic issues observed need to be fixed, as well as several key aspects of the Results section: (1) methods are heavily recapitulated in the Results section, whereas only key aspects of methods should be mentioned in Results and the most important methods information focused on in the Methods section; (2) the Results section is harder to follow because of the high-level of detail dedicated to each pathway reported, as well as support information on each pathway provided in Results. Additional evidence supporting the pathways identified in study that is not directly part of the work performed in the study should be reported in the Discussion section. Finally, while it is understandable that scientific writing in a non-native language is immensely challenging, and the authors have made a genuine effort in their writing to produce a readable article, it would be helpful to have a native or highly-fluent English speaker fix some of the grammatical issues identified as well as help with better word choice where phrasing is ambiguous and leads to potential misunderstandings; this would greatly improve the manuscript with relatively little effort. I have made an attempt to identify grammatical issues or sections that may potentially be misunderstood by readers.

Based on these considerations, it is recommended that the manuscript undergo substantial revision and resubmission for a second evaluation. The content is of sufficient interest to the genetics, feature analysis, and COVID research communities that a revised version of this manuscript would be a valuable addition to the COVID genetics literature.

We sincerely acknowledge reviewer#1 for the thorough evaluation of our manuscript. We are grateful for the constructive criticism that we addressed below. We have thoroughly revised the entire manuscript, including near-complete re-organisation, re-drafting and shortening, with major attempts to clarify the entire paper. We believe that, as a result, it is overall improved.

Major Comments:

1) Formatting: It should be noted that the Nature Communications Biology reference notation is to use superscripted numerals (e.g., “1”), whereas the entirety of the text uses citations formatted in a first author-date format (e.g., “Pairo-Castineira et al., 2020”). This will need to be changed throughout the text.

We have reformatted reference notation style throughout the text according to the Nature Journal style

2) Results: Throughout the Results section, methods are described in great detail rather than focusing exclusively on the salient findings of these analyses. For instance, the cross-validation approach employed is described over several sentences, but the model performance is described only in one sentence. This section should be revised to indicate clearly what the different models found, and the methodological descriptions folded into the Methods section. Only the most important aspects of modeling that are needed for the findings to be understood should be described in the Results section.

We have revised the results section by moving methodological details into the Methods section. See for instance paragraphs “Comparing genetic variation in severe and asymptomatic individuals” (see lines 115-118, 126-129, 134-136, 150-156)

3) Results: This section lists the pathway findings and much background detail, but the abundance of information presented makes it hard to identify the findings of greatest importance. Where tables and figures can provide adequate information on the pathways with most significant associations, it may be useful to shorten the description of these findings and exclude descriptions of background details on all pathways or save those details for the Discussion section. This would tighten up the section and make findings more salient for the readers.

We have considerably shortened the Results section by commenting only the most significant pathways of the largest clusters (e.g. lines 165-246). We still comment a few variants in this section, in particular those affecting the genes (such as *PLEC* or *PCSK5*) that we comment later in the PheWAS section.

4) Methods: the software packages used are listed, but not cited, and it is unclear where certain packages (e.g., “Table2x2”) are implemented, whether in R, scikit-learning, or elsewhere. These details are in the last sentence of methods, but might be clearer if they are reported earlier in the Methods section.

We referenced the libraries throughout the text (see lines 445, 476)

Minor Comments

1) Abstract (page 2, line 39): It is recommended to change “allowed to identify a handful of 16 variants” to “allowed us to identify 16 variants” or “helped us identify 16 variants”. The expression ‘handful of’ does not typically include an actual number, and usually denotes fewer individual items than 10.

We rephrased the sentence as “allowed us to identify 16 variants”.

2) Abstract (page 2, line 42): It is recommended to change “good accuracy” to “high accuracy”, as ‘good’ represents a qualitative description rather than ‘high’ which captures the level of or degree of something.

Done (line 41)

3) Abstract (page 2, line 42): It is recommended to replace “ROC_AUC” with “AUCROC” here, and wherever else is needed in the text.

Done

4) Abstract (page 2, line 42): It is recommended to change “confirming their link with COVID-19 severity outcome” to “supporting their link with COVID-19 severity outcome” or even “strongly supporting”. This is suggested because ‘confirming’ may be perceived as an indication of direct causality, and while there is strong evidence, additional phenotypic modeling is needed to confirm the function of these variants.

We replaced “confirming” with “supporting” (line 51)

5) Introduction (page 3, lines 64-65): It is recommended to change “challenging at an unprecedented level health, economical and societal systems worldwide” to “challenging health, economic, and social systems worldwide at an unprecedented level.”

We rephrased the sentence following the suggestion (lines 61).

6) Introduction (page 3, lines 68): Change “ARSD” to “ARDS” in all instances it is used.

Done

7) Introduction (page 3, lines 75): A citation labeled “1” is missing- please add this in.

We rephrased the sentence “higher body mass index¹” to “higher body mass index” and added a proper reference (<https://doi.org/10.7326/M20-3742>) (line 68).

8) Introduction (page 3, lines 76-79): This is a run on-sentence. It is recommended to consider breaking it in two sentences, where the phrase “...also considered that” would start the new sentence, rewritten as something like “It is important to consider that,...” or “It should also be remembered that,...”

We have split the sentence in two and rephrased it accordingly (lines 70-72).

9) Introduction (page 3, line 81): It is recommended to change “genes involved in type I interferon (IFN)” to “genes involved in type I interferon (IFN) responses.”

Done (line 74).

10) Introduction (page 3, line 83): Remove the word “campaigns.” Alternatively, replace it with “projects.”

Done (line 75).

11) Introduction (page 3, lines 85-86): Replace “Up to now,...” with “Until now,...”

Done (line 78).

12) Introduction (page 4, line 97): Change “While GWAS studies provide solid foundations...” to “While GWAS provide solid evidence...”

Done (line 85).

13) Introduction (page 4, lines 112-113): Change “to identify a few dozens of genetic variants” to “to identify a few dozen genetic variants”.

Done (line 100).

14) Results (page 4, line 117): The Results section indicates that a WES dataset of “germline variants” was used, but according to the prior GEN-COVID paper, somatic cells were used for DNA. Please clarify where it is indicated that germline sequencing was performed.

The bioinformatic pipeline for germline variant calling was used. This has nothing to do with the cell type used for sequencing (blood in this case). The analysis pipeline is based on GATK4 best practices as described in more details in previous papers . We now refer to this methodological details in the Methods section (lines 436-437)

15) Results (page 5, lines 160-161): Change the “F1-score” to “F1” here and “F1” or “F1 score” throughout the text.

Done (line 135).

16) Results (page 6, lines 179-182): Rephrase this sentence to flow more smoothly. For example, “We observed a high level of performance when we tested the ensemble of models trained with only informative variants...” The phrase “achieved good performances” is fine, but it is a qualitative description rather than a level of performance. I would also suggest removing “Remarkably” as it could be seen as editorializing, however this word could stay as it draws attention to the finding.

We rephrased the sentence in the following way: “We observed a high level of performance when we tested the ensemble of models trained with only informative variants on a follow-up cohort of 618 individuals (122 asymptomatic, 496 severe; Fig. 1C), either at the individual model level or at the ensemble one (Table S4)” (lines 154-156)

17) Results (page 6, lines 198-199): Change “associated to cardiomyopathies” to “associated with cardiomyopathies.”

Done (line 173-174).

18) Results (throughout): Scientific notation for decimals are incorrectly reported and should be modified throughout. For instance, the FDR reported for the association of arrhythmogenic right ventricular cardiomyopathy is reported as “FDR=4.03e-05”. In various program outputs the capital letter “E” stands for “Exponential Notation” and the number following it is not superscripted. Using a lowercase “e” and superscripted number indicates raising Euler’s number, the mathematical constant approximately equal to 2.71828, raised to the power shown. So the value “FDR=4.03e-05” has the value FDR=0.027154 rather than

the value the investigators are trying to indicate, which is $FDR=0.0000403$. Throughout the text, wherever it says “e-x”, this should be changed to either “E-X” or “ $\times 10^{-x}$ ” (e.g., report “ $FDR=4.03e-05$ ” instead as “ $FDR=4.03\times 10^{-5}$ ”.

We are sorry for the oversight, thanks for pointing this out. We have now homogenized the scientific notation for decimals, using the “E” capital letter followed by the exponent number not superscripted.

19) Results (page 7, line 211): Change “remarking” to “marking”.

Done (line 181).

20) Results (page 7, line 222): Instead of reporting log Odds Ratios in the text, it is recommended to report them as Odds Ratios (OR) as those are easier to understand. Other than the log-odds ratio, these could be reported as their mathematical equivalent, β s. For instance, a $\log(OR)=-1.34$ is easier to understand if described as an $OR=0.26$, which indicates that the association is highly protective.

We now report all the associations as Odds Ratios.

21) Results (page 7, line 230): Change “All these three pathways are participated by” to “All three pathways include”.

Changed accordingly (lines 199-200).

22) Results (page 7, line 233): Change “top20 of genes” to “top 20 genes”.

Done (now line 203).

23) Results (page 7, lines 236-238): It is unclear what is meant by “rs150021157... affecting the PCSK5 gene” if that means that “rs150021157... located in the PCSK5 gene”. This may be a misreading on my part.

The tool we used for annotation (Annovar; Table S1) and Ensembl (https://www.ensembl.org/Homo_sapiens/Variation/Mappings?db=core;r=9:76327528-76328528;v=rs150021157;vdb=variation;vf=210484809) agree in reporting a missense mutation for several transcripts of the PCSK5 gene as a consequence of the rs150021157 variant. The OpenTargetsGenetics platform also suggests PCSK5 as the gene most affected by this variant (according to the integrated scoring system V2G), however it also reports LFK as the closest gene in terms of distance of the variant from the transcription start site (https://genetics.opentargets.org/variant/9_76328028_G_C). Given this ambiguity, we prefer to generally talk of an influence of the rs150021157 variant on the PCSK5 gene. We have now rephrased this sentence as follows (lines 205-208): “Another variant with 100% support within the same cluster is rs150021157, which is significantly enriched among severe patients ($OR=3.95$; $pval=1.9E-03$; Table S1,S8) and it affects the PCSK5 gene, a serine endoprotease which processes various proteins including cytokines, NGF, renin and which has been reported to regulate the viral life cycle”

24) Results (page 8, line 266): The values “ $pval=0.002881778$ ” should be reduced to three significant figures (“ $pval=0.00288$ ”) in keeping with results reported elsewhere in the paper. All other p-values should be set at three significant figures.

We have homogenized the p-values using the scientific notation as described above.

25) Results (page 8, line 268): Change “which entails CEP131” to “which includes CEP131”. It is unclear whether the statement means that “affected by the variant rs2659015” means that the variant is located in the gene or the variant affects that gene in some way.

In this case, there is an agreement for the various annotations (Annovar: Table S1; Ensembl: https://www.ensembl.org/Homo_sapiens/Variation/Mappings?db=core;r=17:81198884-81199884;v=rs2659015;vdb=variation;vf=104728126 ; OpenTargetsGenetics: https://genetics.opentargets.org/variant/17_81199384_T_C) to locate the variant and the effect inside the CEP131 gene.

26) Results (page 9, line 295): Change “three groups of patients on the original cohort” to “three groups of patients in the original cohort”.

Done (line 253).

27) Results (page 9, line 298): Change “(70%) of the total” to “(70% of the total)”.

Done (line 256).

28) Results (page 10, line 313): Change “associated to disease traits” to “associated with disease traits”.

Done (line 270).

29) Results (page 10, line 344): It is unclear what “invariably relevant for severity classification” means- a different word choice may be helpful.

We have rephrased the sentence in the following way (lines 300-301): “Two of the variants enriched among severe patients which had the highest importance in all our models (i.e. PCSK5 rs150021157 and PLEC rs140300753)”

30) Discussion (page 11, line 379): Change “genetic interactions interplaying with virus genetics” to “genetic interactions with virus genetics”.

Done (line 334).

31) Methods (page 14, line 486): Change “We employed a Log-Odds Ratio (LOR) statistics to perform case-control association” to something like “We used logistic regression to estimate log-Odds Ratios (LORs) for testing association in our case-control datasets”. Without this change, it is less clear how the analysis was done.

We didn’t use a logistic regression model to test case-control associations, but a simple log Odds Ratio calculated on a 2x2 contingency table. So we think it is fair to leave it like that.

Reviewer #2 (Remarks to the Author):

The paper claim to have identified variants with nonlinear interactions associated with severe covid. The main difficulty with this paper is the method of patient selection that uses age and sex to predict the outcome. This will make the analysis prone to cohort effects. Therefore, the differences in variant frequency between cases and controls could be attributed to differences in allele frequencies between older and

younger patients. Hence, all analyses need to be repeated without making the(recursive) selection based on the same sample used to identify variants.

We thank reviewer #2 for his comment which touches on an important point of the analysis. To remove the potential effect of age for the selection of variants, we refined patient classification based on disease phenotype grading to make it concordant with the severity score adjusted by age outputted by a previously developed ordinal logistic model which uses age as input feature for sex-stratified patients (<https://doi.org/10.1007/s00439-021-02397-7>). This led to the removal of 89 patients from grading 0 and 100 patients from grading 3+4+5. We believe that by using this approach we are removing patients that are at higher (or lower) risk on the basis of their age. To assess the effect of the grading classification adjusted by age, we retrained the models by considering all the patients on the basis of the original grading classification (for a total of 1078 patients). The new models never achieved performances as high as the previous ones, both when tested in the set held out from the training on the starting cohort (that we call “2000 cohort”; see Figure A below) as well as when tested in the follow-up cohort of unseen patients(that we call “3000 cohort”; see Figure B below). In particular the models considering variants+covariates or variants only always performed worse when we train the models on patients grouped on the basis of the original grading. This confirms that the refinement of the cohort on the basis of the grading adjusted by age improves the detection of variants that are more predictive of severity.

A

B

- Full supported variants+covariates adjusted grading 3000 cohort AUC = 0.96
- Full supported variants only adjusted grading 3000 cohort AUC = 0.57
- Only covariates AUC adjusted grading 3000 cohort = 0.80
- Full supported variants+covariates unadjusted grading 3000 cohort AUC = 0.88
- Full supported variants only unadjusted 3000 cohort grading AUC = 0.51
- Only covariates AUC unadjusted grading 3000 cohort = 0.88

Reviewers' comments:

Reviewer #1 (Remarks to the Author):

This manuscript describes a whole-exome sequencing (WES) study looking at ~2,000 patients with varying levels of COVID-19 severity in order to identify rare variants that may correlate with severity of disease. The analytic strategy employed by the investigators used multiple search strategies including machine learning approaches like feature importance analysis to identify and rank variants most associated with severity of disease, and then to predict severity based on these variants. The top 16 variants identified by all modeling approaches showed evidence of being involved in severity of the phenotype, including through follow-up PheWAS analyses of the variants which indicated that these same variants were also involved in respiratory diseases and disorders, suggesting biological plausibility for their role in severity of COVID pulmonary symptoms. Predictive modeling of these same genetic variants found a modest improvement in prediction over the predictive ability of a model including just age and gender, AUCROC=0.86 (max AUCROC=0.91) vs. AUCROC=0.80, suggesting that genetic variants are useful in predictive modeling, but not as important as clinical covariates of importance. While this does provide an example of the utility of feature importance analysis, it may overstate the utility of identifying predictive genetic variants in the context of a disease like COVID for which much of the variability in phenotype expression is driven by comorbidities.

Some concerns about age differences between the comparison groups led to a suggestion by the second reviewer to account for age disparities to account for cohort effects (i.e., allele frequencies varying by age), and while the resulting exclusions weakened the model somewhat, the newer findings are consistent with previous findings, arguably strengthening the study findings.

Furthermore, the linguistic/stylistic concerns and the repetitiveness of certain methodological components as identified in the previous review have largely been alleviated, and it is recommended that the manuscript be accepted for publication. As before, the content of this work is of sufficient interest to the genetics, feature analysis, and COVID research communities that a revised version of this manuscript would be a valuable addition to the COVID genetics literature.

Reviewer #2 (Remarks to the Author):

Thank you for letting us see the ROC curves for the different analyses. The sample selection is made by fitting a classifier to the same sample that is used to train the final classifier. The classifier obtained this way achieves a high performance on this subset of samples. I think the analysis does this way does not have out of sample validity, because the same sample is used twice. Can you check the performance of the learned classifier on the samples that were removed in the first step?

Reviewers' comments:

Reviewer #1 (Remarks to the Author):

This manuscript describes a whole-exome sequencing (WES) study looking at ~2,000 patients with varying levels of COVID-19 severity in order to identify rare variants that may correlate with severity of disease. The analytic strategy employed by the investigators used multiple search strategies including machine learning approaches like feature importance analysis to identify and rank variants most associated with severity of disease, and then to predict severity based on these variants. The top 16 variants identified by all modeling approaches showed evidence of being involved in severity of the phenotype, including through follow-up PheWAS analyses of the variants which indicated that these same variants were also involved in respiratory diseases and disorders, suggesting biological plausibility for their role in severity of COVID pulmonary symptoms. Predictive modeling of these same genetic variants found a modest improvement in prediction over the predictive ability of a model including just age and gender, AUCROC=0.86 (max AUCROC=0.91) vs. AUCROC=0.80, suggesting that genetic variants are useful in predictive modeling, but not as important as clinical covariates of importance. While this does provide an example of the utility of feature importance analysis, it may overstate the utility of identifying predictive genetic variants in the context of a disease like COVID for which much of the variability in phenotype expression is driven by comorbidities.

Some concerns about age differences between the comparison groups led to a suggestion by the second reviewer to account for age disparities to account for cohort effects (i.e., allele frequencies varying by age), and while the resulting exclusions weakened the model somewhat, the newer findings are consistent with previous findings, arguably strengthening the study findings.

Furthermore, the linguistic/stylistic concerns and the repetitiveness of certain methodological components as identified in the previous review have largely been alleviated, and it is recommended that the manuscript be accepted for publication. As before, the content of this work is of sufficient interest to the genetics, feature analysis, and COVID research communities that a revised version of this manuscript would be a valuable addition to the COVID genetics literature.

We would like to thank again Reviewer 1 for the extensive revision and thoughtful comments that helped to improve our manuscript.

Reviewer #2 (Remarks to the Author):

Thank you for letting us see the ROC curves for the different analyses. The sample selection is made by fitting a classifier to the same sample that is used to train the final classifier. The classifier obtained this way achieves a high performance on this subset of samples. I think the analysis does this way does not

have out of sample validity, because the same sample is used twice. Can you check the performance of the learned classifier on the samples that were removed in the first step?

We thank Reviewer 2 for the additional comments and suggestions. We have further tested the model on patients that were previously excluded from both the training as well as from the testing set, due to inconsistency between the original WHO grading classification and the one outputted by the ordinal logistic regression classifier adjusted by age developed previously by some of the co-authors (Fellarini et al., Human Genetics, 2022). In details, in the original cohort that we used for training the model, there were 237 samples from either asymptomatic (grading 0) or severe (grading 3+4+5) patients that were excluded due to classification inconsistencies, while in the follow-up cohort used for final testing of the model, 220 more individuals were excluded according to the same criteria. After removing patients with missing values, we obtained an aggregated list of 375 unique, unseen samples never considered in any previous training and testing steps, which we used for further testing the model. Also in this set, our ensemble of models is able to classify severe patients with good accuracy (ACC=85.34%, MCC=67.8%, AUCROC=91.4%).

These results demonstrate that the trained model is able to reliably classify severe patients without any prior classification step using the adjusted by age ordinal logistic regression model.

We have added a supplementary figure (Fig. S1) and table (Table S6) with the performances referred to this second test set and we renumbered all supplementary figures and tables accordingly. We have also commented these new results in the Results (lines 169-172) and Methods section (lines 544-551).